# Intraspecific and intraindividual trait variability decrease with tree richness in a subtropical tree biodiversity experiment

Pablo Castro Sánchez-Bermejo [1,2,3] ✉, Carlos P. Carmona [4,5], Meredith Christine Schuman [6,7], Raquel Benavides [8,9], Lena Sachsenmaier [2,10], Shan Li [11], Xiaojuan Liu [11] & Sylvia Haider [3]

Phenotypic variability within tree species responds to local tree species richness. However, we lack evidence on how different sources of trait variation shape tree-tree interactions. Along a diversity gradient from one to eight tree species, here we collect 4568 leaves from 381 trees to study changes in intraspecific and intraindividual leaf trait variability, and assess their contribution to community functional diversity. Intraspecific trait variability in functional traits decreases with tree species richness, while similar responses for intraindividual variability are revealed by spectral traits. Functional overlap between conspecific trees increases through intraindividual variation, but is reduced through intraspecific variability, meaning that intraspecific variability may reduce intraspecific competitive interactions while intraindividual variability could arise due to varying light within the canopy. Last, intraspecific and intraindividual variability explain high community functional richness and divergence, respectively, especially in mixtures. Our findings emphasize that fine-scale variability influences tree-tree interactions and drive local functional diversity.

Plant trait-based ecology focuses on phenotypic differences as a way to understand ecological and evolutionary processes[1-3]. While the field has typically focused on differences between species, substantial trait variation occurs at different levels of biological organization (among populations, between individuals within the same population, or within individuals[4,5]) which could be important to understand adaptations to the environment[6] and species coexistence[7]. For instance, in response to competition, plants are able to shift the trait expression to adopt a more conservative strategy in the use of resources[8] or to

prevent local competitive exclusion by increasing dissimilarities with other individuals[9,10]. However, while these shifts have been widely studied, less attention has been paid to the variability of traits between individuals within the same population, hereafter referred to as intraspecific trait variability (i.e. the extent of the differences between the trait values of individuals from the same population of a species).

Among others, intraspecific trait variability can be driven by species diversity, as it regulates the probability of local interactions with species of varying identities and, therefore, influences the trait

[1]Martin Luther University Halle-Wittenberg, Institute of Biology/Geobotany and Botanical Garden, Halle (Saale), Germany. [2]German Centre for Integrative Biodiversity Research (iDiv) Halle-Jena-Leipzig, Leipzig, Germany. [3]Leuphana University of Lüneburg, School of Sustainability, Institute of Ecology, Lüneburg, Germany. [4]Misión Biológica de Galicia (MBG), Consejo Superior de Investigaciones Científicas (CSIC), Pontevedra, Spain. [5]Department of Botany, Institute of Ecology and Earth Sciences, University of Tartu, Tartu, Estonia. [6]Department of Geography, Faculty of Science, University of Zurich, Zurich, Switzerland. [7]Department of Chemistry, Faculty of Science, University of Zurich, Zurich, Switzerland. [8]Department of Natural Systems and Resources, ETSI Montes, Forestal y del Medio Natural, Universidad Politécnica de Madrid, Madrid, Spain. [9]Center for the Biodiversity Conservation and Sustainable Development, CBDS-UPM, Madrid, Spain. [10]Systematic Botany and Functional Biodiversity, Leipzig University, Leipzig, Germany. [11]State Key Laboratory of Vegetation and Environmental Change, Institute of Botany, Chinese Academy of Sciences, Beijing, China. ✉e-mail: pablokstrosb@gmail.com

expression of plants[11]. In this context, as the limiting similarity theory suggests that individuals can coexist only if they acquire resources differently[12], intraspecific trait variability within populations may reduce intraspecific competition by allowing individuals from the same species (conspecifics) to exploit alternative resources[13,14]. This suggests that, given that conspecifics acquire and use resources in a similar way, intraspecific trait variability is expected to be larger in species-poor communities. Further, intraspecific trait variability depends not only on intraspecific competition but also on niche availability[15]. That is why, when the number of species in a community increases (which commonly results in resource partitioning[16]), individuals tend to become more dissimilar from the heterospecific neighbors. As a result, species may adopt a so called niche packing strategy characterized by the exploitation of a specific resource in a specific manner, resulting in lower intraspecific trait variability compared to monocultures[17]. For example, conspecific trees in monocultures have been found to produce leaves with different specific leaf areas (SLA) to exploit different sections of the light gradient; by contrast, in mixed communities, as species specialize in exploiting specific parts of the canopy space, conspecifics tended to produce leaves with similar SLA[18]. As a result, limited intraspecific variability may act as a mechanism that would allow species to exploit different niches, resulting in species complementarity in species-rich communities[19]. In fact, recent studies found that intraspecific leaf variability of plant populations decreased with increasing plant diversity[17,20,21]. However, in contrast to the limiting similarity rationale, plant-plant interactions can also be driven by competitive hierarchies, meaning that traits shift towards a more competitive ability depending on the closest neighbors[9]. In this context, individuals from the same species may adjust their traits to varying competitors' identities. Therefore, a diverse community could result in an heterogenous biotic environment in which plants from the same species adopt different strategies simultaneously, resulting in higher intraspecific trait variability compared to monocultures[11]. In fact, this is supported by results from other studies and by the responses found for tree organs other than leaves[22–24]. For instance, Benavides et al.[23] showed that intraspecific trait variability in tree species was higher in mixtures compared to monocultures, specially in relation to architectural traits, and discussed that this change may arise from the spatial complementarity provided by species dissimilarities. As a result, these contrasting results among studies propose that there is no general direction of change of intraspecific trait variability in response to species diversity, but that it likely depends on the specific interaction partners as well as the plant organ studied.

Scaling down in the levels of biological organization, intraindividual variability, i.e., the extent of different trait values across different repeated architectural units of the plant body structure (e.g. leaves from the same plant[25–27]), could also matter for plant-plant interactions and, therefore, may respond to species diversity. For instance, as the light interception by leaves is a key factor in competition[28], plants express different leaf phenotypes within individuals to adjust to light exposure (e.g. leaves directly exposed to sunlight show higher photosynthetic rates than shade leaves[28,29]). Additionally, it has been suggested that plants can experience intraindividual changes in ecophysiological traits that may eventually lead to enhanced water-use efficiency[30] or cope with environmental unpredictability[31]. As species composition affects spatial arrangement and, therefore, light exposure, leaves also respond to the surrounding diversity[17]. This may be relevant in the case of trees as such plastic responses are especially noticeable in these organisms due to their great longevity and extensive modularity[32]. For instance, intraindividual leaf variability in trees was observed to decrease with local tree species diversity[17,33] and it has been suggested that, similarly to intraspecific variability, high intraindividual leaf variability could support intraspecific complementarity (functional complementarity between conspecifics). This would imply

that, under scenarios of high intraindividual variability, conspecifics may tend to be dissimilar in their leaves by exploiting dissimilar niches. Nevertheless, this role of intraindividual variability in plant-plant interactions and the mechanisms involved remain still unclear. Indeed, changes in intraindividual trait variability may not be necessarily related to limiting similarity, but, instead, may respond to spatial and environmental heterogeneity of the environment. For instance, in forests, enhanced intraindividual trait variability emerges in monocultures, where low canopy density and structural diversity result in less buffered environmental conditions[34]. This variability may help to cope with the higher temporal variability in environmental conditions over time[33]. That is why, in order to clarify whether intraindividual variability generates intraspecific complementarity, it is important to understand the patterns of intraindividual variability along tree species diversity gradients as well as its effect on how different conspecifics overlap in their traits.

Functional diversity (i.e. the extent of phenotypic differences in a community) is one of the most common tools in trait-based ecology[35] and can reveal key facets of ecosystem functioning (e.g. net primary productivity, biochemical cycles) and community assembly (e.g. environmental filtering, limiting similarity[1,36]). Functional diversity estimations typically consider a single mean trait value for each species; this strategy reduces the amount of trait measurements but neglects trait variation within species[2]. Nevertheless, intraspecific trait variability can account for a non-negligible proportion of the total trait variability within and across ecological communities[4,37]. Further, approaches considering intraindividual trait variability have shown that the sum of the variation occurring intraspecifically and intraindividually may be even larger than the differences between species in the case of some leaf traits such as SLA or leaf nitrogen content[5,38]. This shows the importance of studying species traits beyond single mean trait values to quantify community functional diversity, especially at local scales and in species-poor communities[39–42]. Therefore, it has been suggested that the different sources of trait variation occurring within species, from the variability between populations to the intraindividual variability, could affect community functional diversity[43]. In recent years, different methods to incorporate variability into functional diversity metrics have been developed[39,44–47]. The use of these methods allows testing the notion that community functional diversity is higher when considering intraspecific or intraindividual variability[43,48], as well as understanding how this effect changes with species richness.

Here, we study the patterns of intraspecific and intraindividual leaf trait variability in a tree diversity experiment in subtropical China (BEF-China[49]). The modular architecture of trees enables pronounced plastic responses[50], making this system particularly suitable for exploring how trait variability changes with increasing species richness (see proposed hypothesis in Supplementary Fig. 1). By using leaf spectroscopy, we estimate five morphological and chemical leaf functional traits in 381 tree individuals from eight species along a tree species richness gradient with monocultures and mixtures of 2, 4 and 8 tree species. Additionally, we identify 29 leaf spectral traits associated to different segments of the leaf reflectance spectrum. We assess population variability in functional and spectral traits by using two functional indices that measure different facets of the functional space (functional richness and functional divergence) at the intraspecific (mean trait values of individual trees within the same population) and the intraindividual level (leaf trait values within an individual tree), respectively. Further, we assess intraspecific overlap as the shared trait space between the trait distributions of conspecific trees belonging to the same population. Specifically, we aim (1) to determine how tree species richness affects intraspecific and intraindividual leaf functional and spectral trait variability, and (2) to assess the direct and indirect effects (via intraspecific and intraindividual variability) of tree species richness on intraspecific trait overlap. Further, we use a framework

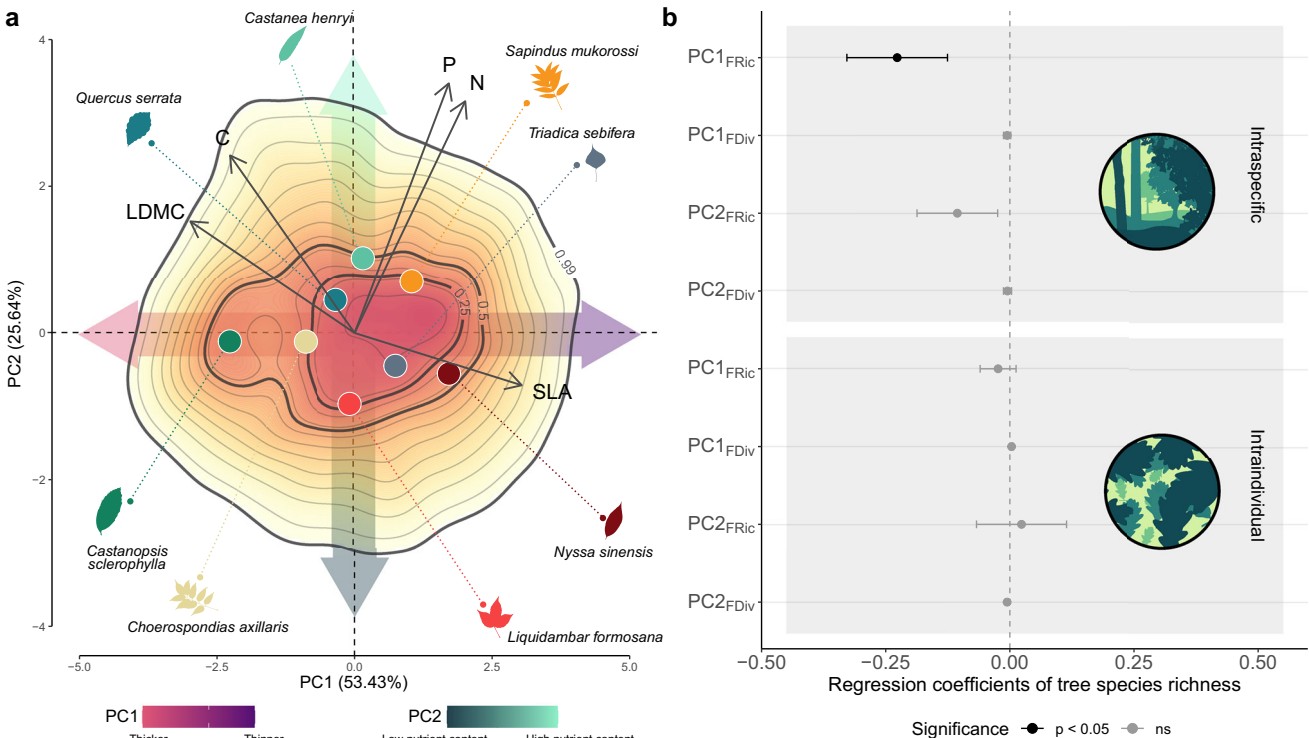

**Fig. 1 | Main axes of leaf trait variation and effect of tree species richness on intraspecific and intraindividual leaf trait variability. a** Biplot for the first two axes of a principal component analysis (PCA) of five functional traits predicted for leaves collected in eight species growing in a subtropical tree diversity experiment (N = 4568; colored points: mean species values). Data are based on spectroscopically predicted trait values of leaves collected from trees growing along an experimental species richness gradient with mixtures of 1, 2, 4 and 8 tree species. The color gradient visualizes different probability densities, with red colors corresponding to portions of the space with highest densities of observations. The first component (PC1) reflects a gradient from thicker leaves (towards the left) that are expected to have a longer lifespan and higher survival probability in response to abiotic and biotic hazards than cheaply constructed thinner leaves (towards the right) which are expected to have higher photosynthetic rates (Supplementary

Figs. 3, 4). The second component (PC2) reflects a gradient in nutrition status ranging from low nutrient content leaves with low photosynthetic capacity (towards the bottom) to high nutrient content leaves that could show high photosynthetic rates. Colors and leaf silhouettes correspond to the tree species included in the study (Supplementary Table 3). Linear mixed-effects models to study intraspecific and intraindividual variability of PC1 and PC2 (**b**) show significant decrease in intraspecific functional richness of PC1 with increasing tree species richness ($\chi^2$(df = 1) = 4.85, $P$ = 0.03, standard estimate ($\beta$) = −0.28). Regression coefficients (points) and standard error (error bars) are shown for the intraspecific level (top, N = 63) and intraindividual level (bottom, N = 381). Colors represent the significance determined by two-sided likelihood ratio test (black $P$ < 0.05, grey $P$ > 0.05).

that allows including hierarchical sources of trait variation on community functional diversity, from the population level to the leaf level, passing through the individual level[39,51], and null models to identify which sources of variation within species affect functional diversity. With this, we aim (3) to characterize the influence of intraspecific and intraindividual variability on the functional diversity of a community across levels of tree species richness. The results show that intraspecific trait variability decreases with tree species richness, reflecting a spectrum from high intraspecific complementarity in monocultures to low intraspecific complementarity in mixed communities. The decrease in intraindividual functional trait variability with increasing tree species richness is less evident and does not result in conspecifics adopting alternative trait strategies. However, this decrease in intraindividual variability is prominent in the case of spectral traits. Last, both intraspecific and intraindividual trait variability contribute to functional diversity of the community, especially in the mixtures, but each source of trait variation affects different aspects of the trait space.

## Results

### Responses of trait variability to tree species richness

The first two axes of a principal component analysis (PCA) on the leaf-level values of five functional traits (N = 4568) explained 79% of the total variation in our dataset (Fig. 1a). PC1 was strongly associated with

leaf dry matter content (LDMC), specific leaf area (SLA) and leaf carbon content (C) (with loadings 0.90, −0.89 and −0.67, respectively; Supplementary Table 1) and reflected differences in the content of dry matter from conservative thicker leaves to acquisitive thinner leaves. PC2 was related to leaf phosphorous content (P) and leaf nitrogen content (N) (with loadings 0.70 and 0.65, respectively) and reflected differences in nutrition status.

Overall, intraspecific leaf functional trait variability between individuals within populations decreased with tree species richness. First, analyses with single axes of leaf functional trait variation, which aimed to detect changes associated with specific axis of leaf functional trait variation, revealed that intraspecific variability in PC1 decreased with tree species richness ($\chi^2$(df = 1) = 4.85, $P$ = 0.03, standard estimate ($\beta$) = −0.28, N = 63 for functional richness (FRic; the extent of the functional volume of the population); Fig. 1b, and Supplementary Table 2). However, this effect was not significant for the intraspecific variability of PC2. Second, we estimated trait probability densities based on both PC1 and PC2 (multivariate FRic) to assess the main changes in the total trait space of the population of the population (i.e. between the conspecific trees within each plot). This analysis revealed that tree species richness also had a significant negative effect on multivariate FRic at the intraspecific level ($\chi^2$(df = 1) = 4.60, $P$ = 0.03, $\beta$ = −0.30, N = 63; Fig. 2a). In contrast, we found no effect of tree species richness on multivariate functional divergence (FDiv; the degree

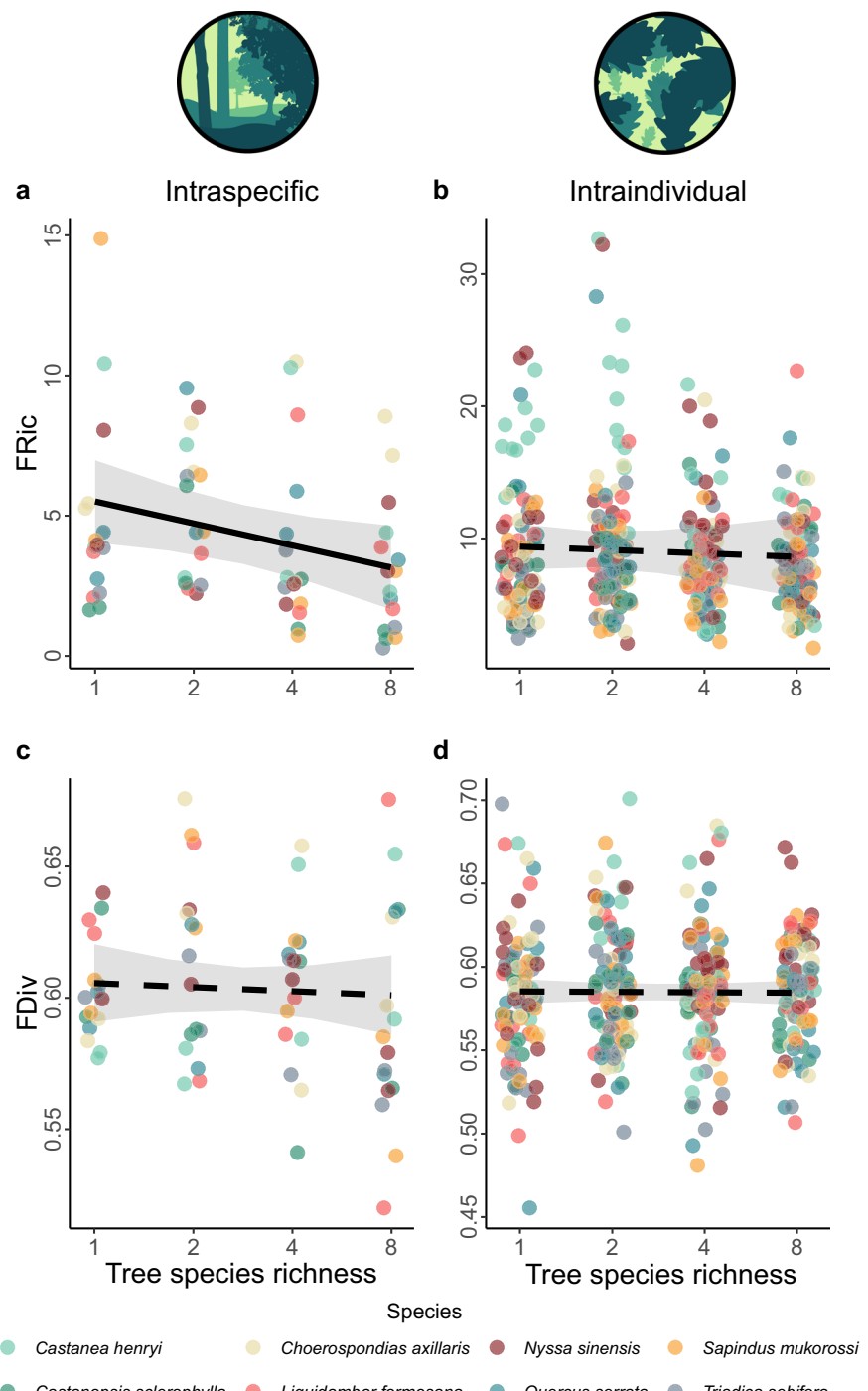

**Fig. 2 | Effect of tree species richness on intraspecific and intraindividual leaf functional trait variability, as reflected by two multivariate functional indices.** Lines correspond to the results of linear mixed-effects models that show (**a**) a significant decrease of intraspecific functional richness (FRic) with increasing tree species richness ($\chi^2$(df = 1) = 4.60, $P$ = 0.047, standard estimate ($\beta$) = −0.30, N = 63) and non-significant effects of tree species richness on (**c**) intraspecific functional divergence (FDiv) and (**b**, **d**) intraindividual FRic and FDiv (N = 381). Data are based on measurements of five morphological and chemical leaf traits in an experimental species richness gradient with monocultures and mixtures of 2, 4 and 8 tree species. Significance was tested using a two-sided likelihood ratio test against a model with no tree species richness effect. Grey bands represent a 95% confidence interval. Colors correspond to the different tree species included in the study (Supplementary Table 3), whose identity was included as a random effect in our models.

to which the abundance in the trait space is distributed towards the extremes of the functional volume) ($\chi^2$(df = 1) = 0.14, $P$ = 0.70, $\beta$ = −0.05; Fig. 2c). The results for intraspecific variability contrast with the effects found for the intraindividual level, as only for intraindividual FDiv of PC2 we found a marginally significant decrease with tree species richness ($\chi^2$(df = 1) = 2.74, $P$ = 0.09, $\beta$ = −0.09; Supplementary Fig. 2c). We found no effect of tree species richness on any of the

multivariate functional indices used at the intraindividual level (FRic and FDiv with $\chi^2$(df = 1) = 0.22, $P$ = 0.64, $\beta$ = −0.06 and $\chi^2$(df = 1) = 0.88, $P$ = 0.88, $\beta$ = −0.01, respectively, N = 381; Fig. 2b, d).

**Responses of spectral variability to tree species richness**

Analyses on leaf reflectance spectra measured on the range of the solar radiation (400-2500 nm) support a decrease of leaf spectral trait

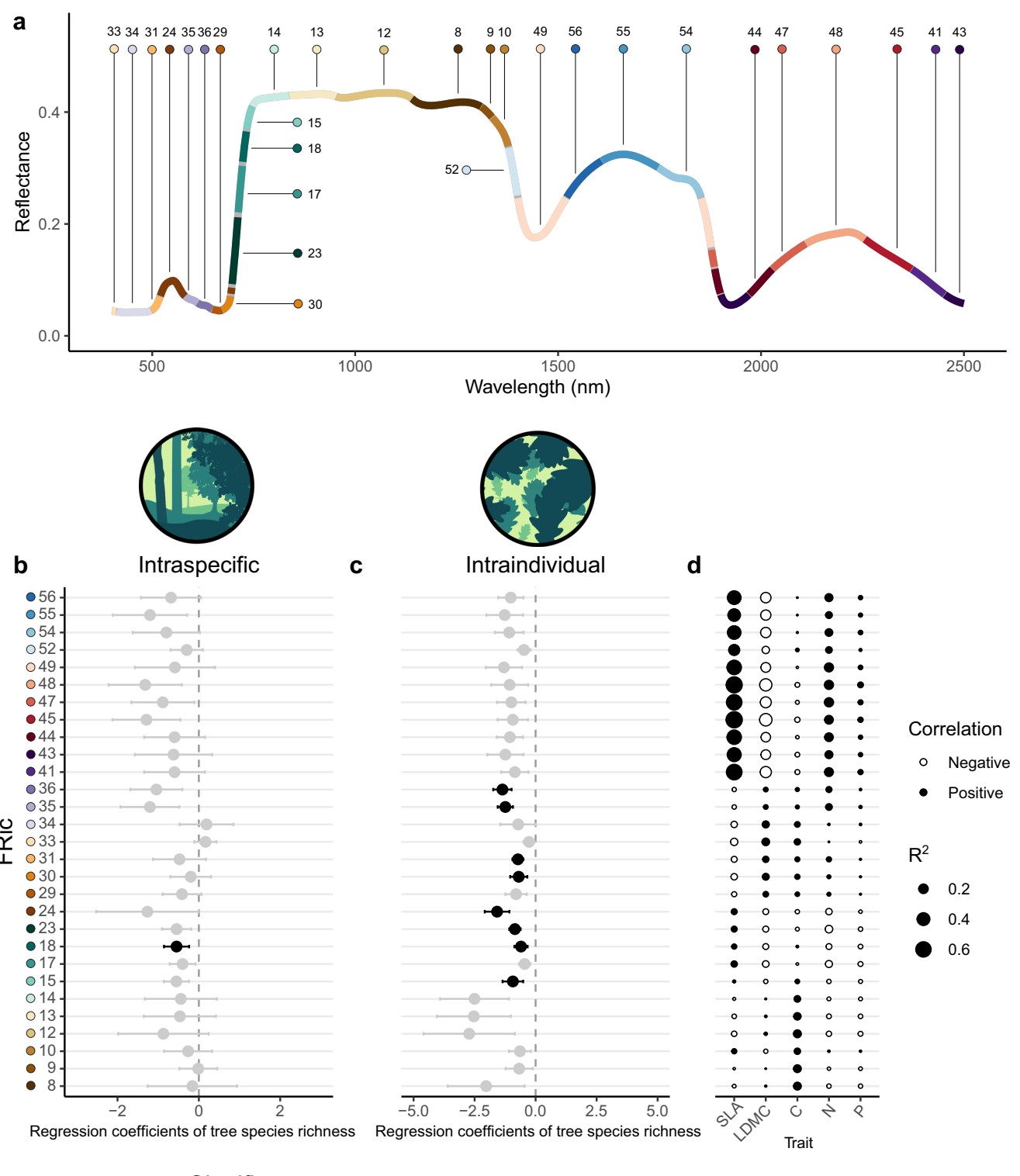

variability with tree species richness, both intraspecifically and intraindividually. Specifically, we found that intraindividual variability (measured as FRic) decreased with tree species richness in the case of 8 principal components associated with different segments of the leaf reflectance spectrum (N = 381; Fig. 3b; see Supplementary Table 4 for P values). For one of these principal components, we also found a significant decrease of FRic at the intraspecific level ($\chi^2$(df = 1) = 4.13, P = 0.04, β = −0.29, N = 63; Fig. 3c). While significant effects were found in the case of FRic, the results for FDiv were not significant (see

Supplementary Fig. 5). The principal components for which we found a response in their variability were associated to segments ranging from 498 to 746 nm (Fig. 3a) and showed low correlations with leaf functional traits (Fig. 3d).

**Effects of trait variability on intraspecific overlap**

We used a structural equation model (SEM) to understand the relationships leading conspecific trees to overlap in the functional trait space of their population (Supplementary Fig. 8, 9). Our model fit the

**Fig. 3 | Spectral segmentation and effect of tree species richness on the variability of spectral components at the intraspecific and intraindividual level.**
**a** Fragmentation of leaf reflectance spectrum into segments holding one identified principal component using a Hierarchical Spectral Clustering with Parallel Analyses (HPS-CA; Supplementary Figs. 6, 7) on 4568 leaf reflectance spectra collected from trees growing along an experimental species richness gradient with mixtures of 1, 2, 4 and 8 tree species. Each segment is represented in a different color and the line represents the mean reflectance measured at different wavelengths (from 400 to 2500 nm) in in our study. Regression estimates from linear mixed-effects to study intraindividual and intraspecific spectral variability of the principal components associated to the identified segments (**b**) show a significant decrease of intraindividual FRic with tree species richness in 8 principal components (the ones associated with segments 15, 18, 23, 24, 30, 31, 35, and 36; N = 381) and (**c**) a significant decrease of intraspecific FRic with tree species richness in one principal component (the one associated with the segment 18). Colors of the regression estimates represent the significance as determined by a two-sided likelihood ratio test against a model with no tree species richness effect (black $P < 0.05$, grey $P > 0.05$), error bars represent to the standard error and colors on the axis correspond to the segments in (**a**). $P$ values and standard estimates for each segment are shown in Supplementary Table 4. **d** Correlation between principal components associated to identified segments and leaf functional traits used in the study. Each circle represents a correlation, with size representing the $R^2$ and the color indicating the direction of change (white, negative; black, positive). Segment numbers are derived from the HSC-PA process shown in Supplementary Fig. 7.

data well (Fisher's C(df = 2) = 0.20 $P$ = 0.91, N = 63). We found that changes in the intraspecific overlap (the mean overlap between the functional volumes of conspecific trees within a population; Fig. 4) are well explained by tree species richness and multivariate intraspecific and intraindividual FRic (marginal $R^2$ = 0.60, conditional $R^2$ = 0.70). Intraspecific FRic significantly decreased with tree species richness (F(df = 1) = 5.10, $P$ = 0.03, β = −0.29) and had in turn a negative impact on intraspecific overlap (F(df = 1) = 78.24, $P$ < 0.001, β = −0.69). However, we did not find any effect of tree species richness on intraindividual FRic (F(df = 1) = 0.36, $P$ = 0.58, β = −0.12), and we found an increase of intraspecific overlap with intraindividual FRic (F(df = 1) = 15.76, $P$ < 0.001, β = 0.33). Additionally, tree species richness had a direct marginal positive effect on intraspecific overlap (F(df = 1) = 2.81, $P$ = 0.09, β = 0.13). These results remained qualitatively similar in SEMs with functional indices based on single axes of trait variation (PC1 or PC2 of Fig. 1a; and Supplementary Fig. 10).

**Effects of trait variation on community functional diversity**
In order to study the importance of intraspecific and intraindividual trait variability in the assessment of functional diversity of a community and its dependence on species richness, we built four null models that randomized functional trait probability densities at three different levels where trait variation arises (Fig. 5, and Supplementary Fig. 11, 12). This approach ensured that simulated assemblages had identical tree species composition as the observed communities, but differed in the functional trait variability within species. Specifically, the sources of random functional trait variation differed between the four null models: (1) random population model (assuming random functional trait distribution of the populations in an assemblage, but within the constraints of the species to which each population belongs), (2) random tree model (assuming random functional trait distribution among trees from the same population, but within the constraints of the species to which each tree belongs), (3) random leaf model (assuming random functional trait distribution among leaves from the same tree, but within the constraints of the species to which each tree belongs) and (4) population-restricted random leaf model (assuming random functional trait distribution among leaves from the same tree, but within the constraints of the population to which each tree belongs; see methods for details on the null models). Based on 500 simulations we calculated the standardized effect sizes (SESs) of FRic and FDiv for every type of null model and every sampled community to determine how much the observed functional diversity deviates from what would be expected under the null models. We then used linear mixed-effects models to study differences in SESs among null models and along a gradient of tree species richness (Supplementary Fig. 12).

We found a significant interaction between tree species richness and the type of model on SES$_{FRic}$ ($\chi^2$(df = 3) = 14.48, $P$ < 0.001, N = 128; Supplementary Table 5). Specifically, SES$_{FRic}$ did not differ from 0 in the random population model, suggesting no differences between the FRic of null models and observed communities. Still, SES$_{FRic}$ became significantly higher than 0 with increasing tree species richness in the random tree null model and the population-restricted random leaf

model, suggesting that FRic in the diverse observed communities was higher compared to FRic from the null models. In the case of the random leaf model, SES$_{FRic}$ was lower than 0 with low tree species richness and similar to 0 in the highest levels of tree species richness, suggesting that in monocultures, observed SES$_{FRic}$ values were lower compared to the null model, and there were no differences between the null model and observed FRic in diverse communities. The interaction between tree species richness and null model was not significant for SES$_{FDiv}$ ($\chi^2$(df = 3) = 4.50, $P$ = 0.21, N = 128), but we found significant effects of tree species richness ($\chi^2$(df = 1) = 18.53, $P$ < 0.001, β = 0.13) and the type of null model ($\chi^2$(df = 3) = 70.56, $P$ < 0.001). For this functional index the random population and the random tree models did not differ from 0 and only the random leaf and population-restricted random leaf models were significantly higher than 0, specially in more diverse communities, suggesting that only for these two last null models the observed values of FDiv were higher than the ones from the null models. All analyses remained qualitatively similar when studying functional indices on single axes of trait variation (PC1 or PC2 of Fig. 1a; and Supplementary Fig. 13).

## Discussion
With our study, we show that intraspecific leaf functional trait variability correlated negatively with tree species richness and in turn, had a strong negative correlation with intraspecific trait overlap within a community. We interpret this to indicate that trees of a given species are on average functionally more similar in species-rich communities than when they are growing in monoculture. In contrast, intraindividual leaf functional trait variability was weakly correlated with tree species richness, but strongly and positively correlated with intraspecific trait overlap. We interpret that, as the leaves within each tree become functionally more similar, the trait expressions of individual trees become more dissimilar to each other. These results are supported by the spectral analyses, which suggest that the decrease in phenotypic variability (at the intraspecific and intraindividual level) is also reflected by spectral traits. Our results also show that the organization of intraspecific and intraindividual functional trait variability influences community functional diversity, especially at higher levels of tree species richness.

Our approach allowed us to study how functional trait variability responded negatively to tree species richness in terms of functional variation between and within individuals. The negative association of tree species richness with functional variability between individuals of a species is consistent with the limiting similarity hypothesis, suggesting that higher intraspecific variability would minimize intraspecific competitive interactions in monocultures, while intraspecific variability is of secondary importance for species coexistence in species-rich communities[13,17,52,53]. Indeed, responses of intraspecific variability in leaf functional traits were found to decrease with species richness in observational studies[21], and also in other BEF-experiments[18], supporting the idea that leaf variability between conspecifics is a mechanism for complementarity in trees. However, these results contrast with previous

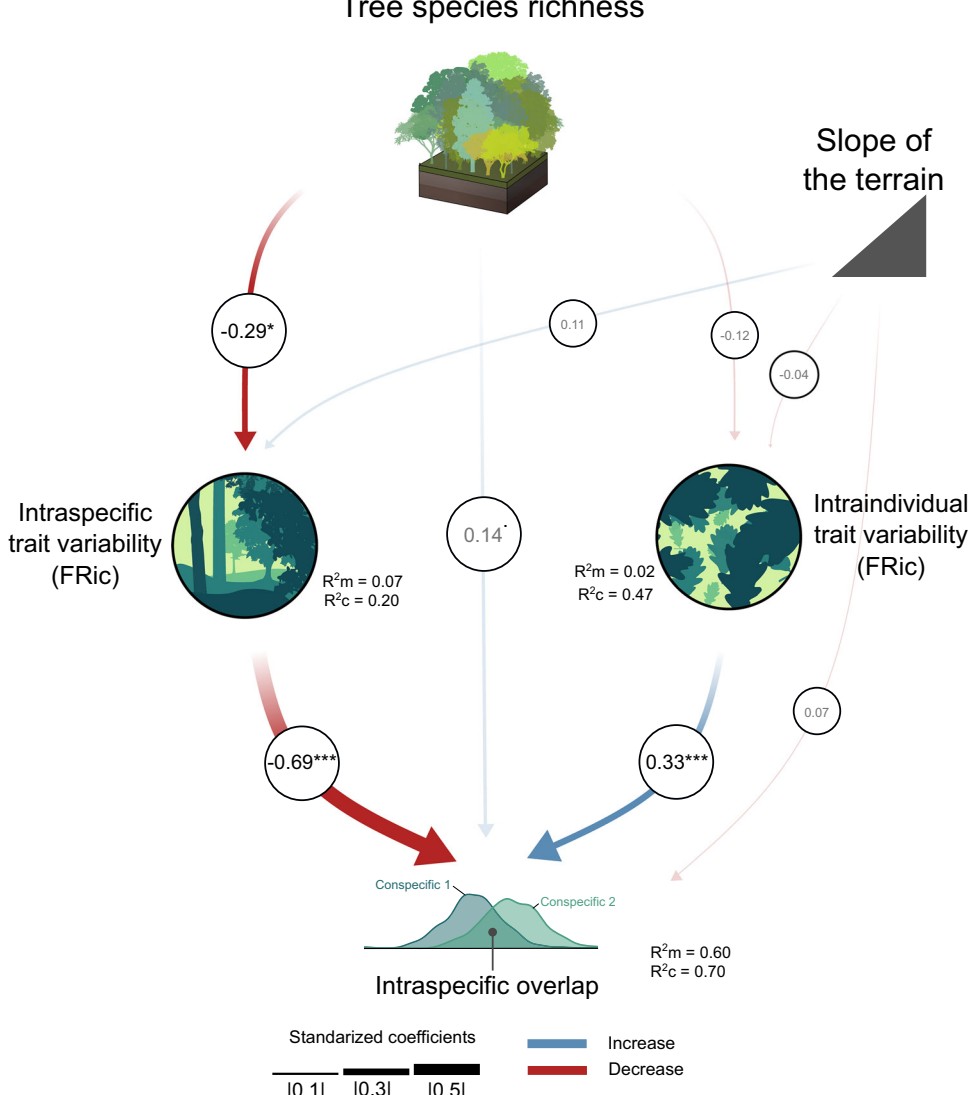

**Fig. 4 | Piecewise structural equation model (SEM) studying the mechanisms driving the intraspecific overlap in leaf functional traits.** The SEM tests the direct effect of tree species richness on intraspecific overlap as well as its indirect effects mediated via multivariate intraspecific and intraindividual variability, which is expressed as functional richness (FRic) here, but see Supplementary Fig. 10 for a non-simplified SEM in which intraspecific and intraindividual functional divergence (FDiv) were also included. Data are based on multivariate functional indices measured at the intraspecific and intraindividual level for eight tree species growing along an experimental species richness gradient with monocultures and mixtures of 2, 4 and 8 tree species. Significant effects of tree species richness were found on intraspecific trait variability (FRic) (two-sided F-test: F(df = 1) = 5.10, P = 0.03, β = −0.29), while intraspecific overlap responded significantly to intraspecific trait variability (FRic) (two-sided F-test: F(df = 1) = 78.24, P < 0.001, β = −0.69) and intraindividual trait variability (FRic) (two-sided F-test: F(df = 1) = 0.36, P = 0.58, β = −0.12). The width and color of the arrows indicate the strength and direction of the effect, with blue arrows showing positive effects and red arrows negative ones. Significant results are represented by solid lines while non-significant relationships are represented by semi-transparent lines. Asterisks indicate significant effects (*p < 0.05, ***p < 0.001), while the dot represents marginally significant effects (p < 0.10). The marginal and conditional R² (R²m and R²c, respectively) are indicated for every model of the piecewise SEM.

observational studies on trees that found an increase in intraspecific functional trait variability with tree species richness[22–24]. These studies mentioned that higher structural complexity (i.e. the structural diversity in the occupancy of the aboveground space) could release competition, allowing species to occupy a larger niche space. In fact, most of the responses found in these studies involved architectural traits (e.g. crown projection area), for which increasing tree species richness often leads to higher complexity in canopy space-filling[54,55]. However, most of these studies were observational and included trees differing in age and distance from neighbors. Such heterogeneous settings would impede, for instance, separating the variability arising from neighborhood diversity from that associated to ontogeny[56,57].

Our results also indicate that variability in the leaf economics spectrum (LES[58]), accounts for most of the changes at the intraspecific level (as indicated by the results of functional richness for PC1). Therefore, most of the variability occurs between conservative leaves which are expected to have a long lifespan and high resistance against abiotic and biotic hazards, and acquisitive leaves with short lifespans which are expected to be fast in the acquisition of resources and efficient in photosynthetic activity[59,60]. This pattern is consistent with the responses found in other studies for SLA[18], a trait widely used as a proxy for acquisitiveness. It suggests that conspecifics adjust their leaf design in terms of resource use (some individuals more conservative and some more acquisitive), resulting in intraspecific coexistence. Consistently, we found a trend towards a positive effect of tree species

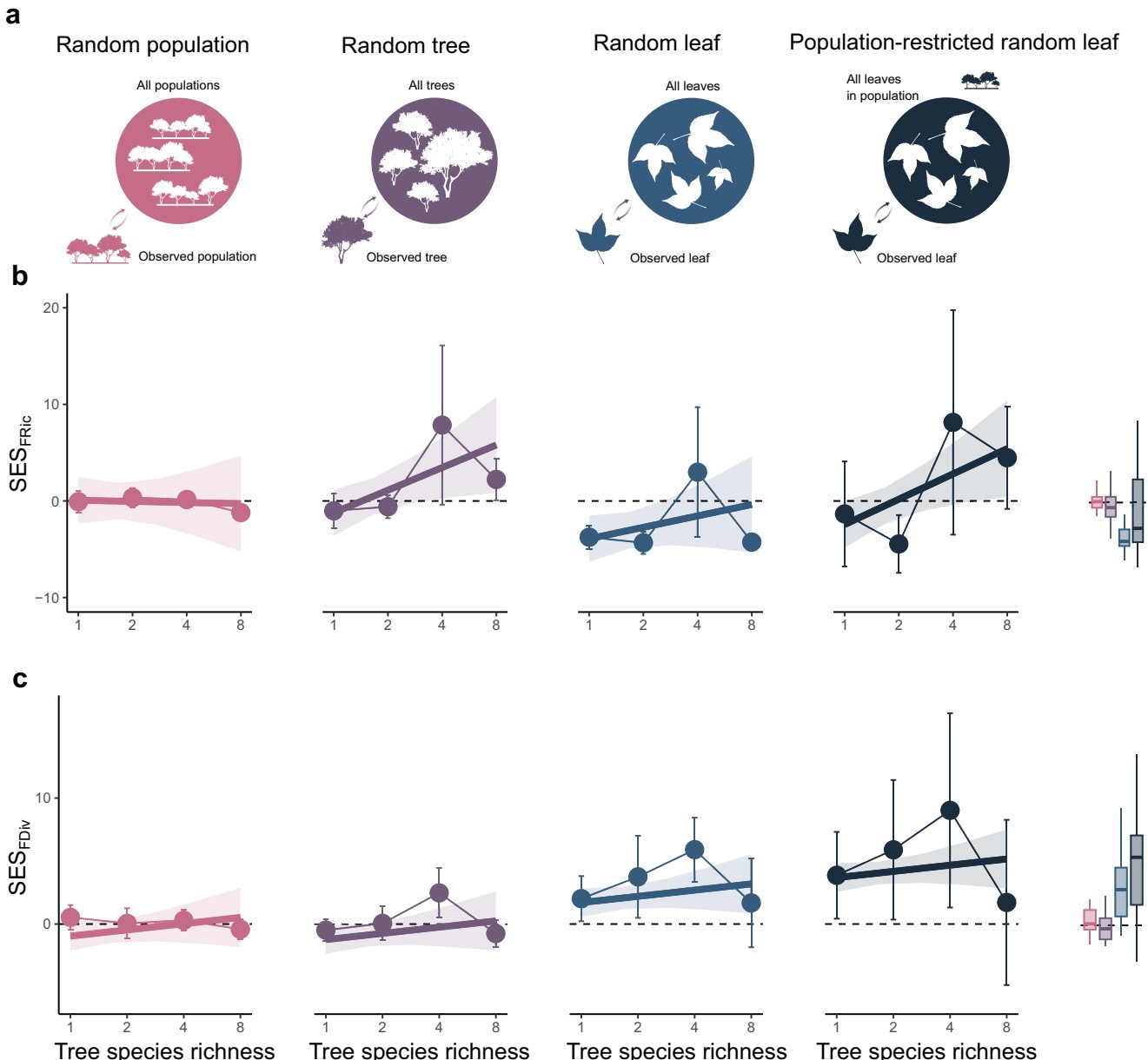

**Fig. 5 | Differences between observed and random values of community functional diversity along a species richness gradient for four null models that randomized different sources of trait variation.** We built (**a**) null models that differ in the level of biological organization in which the randomization was performed (population level, tree level or leaf level; Supplementary Fig. 12) based on leaf-level data collected from plots with 1, 2, 4 or 8 tree species. Data are based on standardized effect sizes (SES) assessed for every null model and functional index (functional richness (FRic) and functional divergence (FDiv)). Linear mixed-effects models showed that the responses of (**b**) $SES_{FRic}$ to tree species richness depended on the type of null model ($\chi^2(df = 3) = 14.48$, $P < 0.001$, N = 120), but (**c**) this interaction was not significant in the case of $SES_{FDiv}$ ($\chi^2(df = 3) = 4.50$, $P = 0.21$, N = 128). However, $SES_{FDiv}$ responded significantly to tree species richness ($\chi^2(df = 1) = 18.53$,

$P < 0.001$, standard estimate (β) = 0.13) and the type of null model ($\chi^2(df = 3) = 70.56$, $P < 0.001$). Significance was tested by using a two-sided likelihood ratio test. SESs are lower than zero (below the dashed line) when the observed values of functional diversity are lower than the simulated ones, while SESs are higher than zero when the observed values are higher than the simulated ones. Semi-transparent bands represent a 95% confidence interval. Points correspond to the mean value of SESs for each diversity level and error bars represent their standard deviation. Boxplots for comparing the values for the SESs in different models are included in the right panels. Horizontal lines inside the box indicate the median, box limits represent the upper and lower quartiles and the whiskers are 1.5 times interquartile range.

richness on intraspecific functional trait overlap. These results reinforce the idea that species richness leads to the convergence of conspecifics in their leaf phenotypic space, resulting in higher niche packing of species due to the higher similarity between conspecifics in the resource-use strategy.

Recent studies suggest that intraindividual variability, which primarily relies on epigenetics and phenotypic plasticity[25], has evolved in natural populations in order to adapt to changing environmental

conditions[30,31,61]. However, this source of variation has been widely disregarded and its role in the context of tree-tree interactions remains largely unclear. Proβ et al.[18] suggested that intraindividual variability in leaves could act in a similar way as intraspecific variability, meaning that higher intraindividual variability could minimize competitive interactions among conspecifics. Therefore, intraindividual trait variability should be higher in monocultures compared to mixtures. However, while a clear decrease in intraindividual variability in

response to tree species richness previously reported, these studies did not explore how this could be related to intraspecific functional trait overlap[33,62]. In contrast, our results indicate that intraindividual leaf functional trait variability promotes trait overlap within a population, suggesting that the role of intraindividual variability in intraspecific complementarity could have been overestimated. Further, while intraindividual variability barely responded to tree species richness in the case of the leaf functional traits used in this study, this effect was prominent in the analyses of spectral traits. In addition, the principal components responding to tree species richness in the case of spectral intraindividual variability were poorly correlated with functional traits from the leaf economics spectrum. This suggests that while traits from the leaf economics provide a better mechanistic and ecological interpretation, they may not be sufficient to understand the patterns of intraindividual trait variability and, therefore, incorporating traits beyond the ones commonly used, as well as other sources of phenotypic information (e.g. leaf spectroscopy), can complement and support the study of phenotypic variability[63]. Interestingly, our results indicate that tree species richness affects intraindividual variability associated to optical properties in the visible range (VIS; 400-700 nm) and the red edge transition (680-750 nm). These regions are strongly linked to leaf photosynthetic and protective pigments, such as carotenoids or chlorophyll[64] (Supplementary Fig. 14), suggesting that the content of these pigments may be more variable in the monocultures compared to the mixtures. As a possible explanation, the lack of clear stratification of the canopy in monocultures[54] could result in a higher differentiation in light availability between sun and shade leaves[32], resulting in higher variability in the content of pigments between leaves of the same tree. In sum, our data suggest that intraindividual variability in tree-tree interactions is associated with intraspecific overlap and may arise as a response to varying light within the canopy.

Our results, interpreted in terms of the deviation of observed functional diversity from null models, showed that the structure of intraspecific and intraindividual functional trait variability contributes positively to communities' functional diversity. While differences among species are still the most prominent source of functional trait variation, intraspecific and intraindividual variability can represent almost half of the total leaf trait variability in our species, especially in the case of predicted leaf nitrogen and phosphorus contents[38] (Supplementary Fig. 15). Therefore, it becomes reasonable that the variability within species will also partly explain how functional diversity is distributed[65]. Surprisingly, although a higher contribution of functional trait variability in monocultures would be aligned with the limiting similarity hypothesis, the increase of the divergence from the null models suggests that intraspecific and intraindividual functional trait variability contributed more to functional diversity in species-rich communities. One possible explanation is that trees tend to differ in trait values from the other trees in the community (from the same or a different species). As a result, this would lead to an increase in the functional diversity of the community that would be more noticeable as the number of species increases[21].

Differences between the deviations in different null models revealed that intraspecific and intraindividual functional trait variability contribute to different facets of functional diversity. In the case of intraindividual variability, the negative departure of FRic from the random leaf null model indicate that, as different populations are exposed to different environmental conditions, the leaves belonging to the same population are highly similar among them compared to other leaves from the same species. Further, the coincidence in the discrepancies between the observed and expected FRic in the random tree and the population-restricted random leaf null models may mean that higher FRic in observed communities is only attributable to intraspecific functional trait variability. In contrast, the higher observed functional divergence (FDiv) in comparison to the expectations of the random leaf and the population-restricted random leaf null

models suggests that communities have a more multimodal distributions, that is, there are several modes (peaks) across the functional space, resulting from intraindividual functional trait variability. Indeed, while the role of intraspecific variability in producing multimodal trait distributions had already been studied[66], our results indicated that this effect could be amplified when considering intraindividual variability. This means that, even within one experimental plot, there is not a unique optimal trait value, but different optimal leaf designs are expressed. This is consistent with previous literature, as due to microenvironmental conditions along the tree crown, multiple leaf designs can be expressed in order to maximize fitness[67], affecting the distribution of traits in the community. Interestingly, while we found differences between the observed functional diversity and the expectations in the random tree, random leaf and population-restricted random leaf model, observed populations did not depart significantly from randomly chosen populations for either functional richness or functional divergence. This suggests that, despite the responses found for intraspecific trait overlap, population identity does not matter for functional diversity. However, we should be careful when interpreting this result because, while we did not find differences in the contributions of different populations from the experiment presented here, populations in natural systems with higher environmental heterogeneity could differ substantially in their contribution to the functional diversity of the community[46].

In summary, both intraspecific and intraindividual functional trait variability responded to tree species richness and affected the distribution of functional diversity. This outcome provides a better understanding of how the variation within species influences functional diversity and supports the idea that intraspecific variability is an important component to be considered when studying the functional diversity of ecological communities at fine or local scales[42,66]. Additionally, we show that intraindividual variability does not only matter for the ecological processes occurring at the population level[68], but also shapes the trait distribution of ecological communities. However, our approach based on the prediction of leaf functional traits from spectral data, while remaining of interest as it allows processing a large dataset that accounts for the intraindividual variability working at a community scale, may raise some concerns. Specifically, although the accuracy of our method to predict some traits for each species individually is high (see Supplementary Table 6), the accuracy decreases for some other traits in some species (e.g. leaf phosphorus content of *Choerospondias axillaris*). This increase in the noise in our metrics of intraspecific and intraindividual trait variability, which in turn may obscure or distort relationships between diversity and trait variation[69]. While general patterns of intraspecific and intraindividual trait variability are supported by direct analyses on the leaf spectral data, results involving traits with lower predictive accuracy should be interpreted with caution, and the absence of significant effects may also reflect methodological constraints next to biological absence of patterns.

Using a trait dataset that accounts for hierarchical sources of trait variation for eight tree species across a gradient of tree species richness, we showed that trait variability within and between individual trees is relevant for understanding patterns of intraspecific functional diversity. Traits are a response to pressures from the abiotic and biotic environment, but simultaneously affect ecosystem functioning[70]. For instance, intraspecific variability in trees has also been shown to be an important factor increasing primary productivity[71], and similar effects are expected for intraindividual trait variability[6]. Therefore, understanding the patterns of trait variation could reveal new facets of the mechanisms behind ecosystem functioning. Altogether, our study demonstrates the importance of considering biological units below the population or species level in trait-based ecology, thus highlighting the importance of moving from a species-based trait ecology to an individual-based trait ecology, that could enable better understanding of processes occurring at the local scales.

## Methods

### Study site and experimental design

This study was conducted in a biodiversity–ecosystem functioning (BEF) experiment, the BEF-China tree diversity experiment, located in Xingangshan, in Jiangxi Province, China (lat. 29°08′11″N, long. 117°90′93″E). While BEF-China was primarily created to investigate ecosystem functions in planted areas with different levels of tree-species diversity, thereby simulating the impact of species extinction, this experiment has also been used to address intraspecific changes of trees in response to their biotic context[72]. The climate is subtropical with a mean annual temperature of 16.5 °C (ranging from 0.4 °C in January to 34.2 °C in July) and mean annual precipitation of 1,821 mm[73]. We worked on Site A, where trees were planted in 2009 and which extends over an area of 27 ha with an elevation ranging from 205 to 275 m a.s.l. and slopes from 8.5° to 40° (Supplementary Fig. 16). In each plot, 400 saplings were planted in a uniform grid with 1.29-meter spacing, resulting in plots of 25.8 by 25.8 meters with species randomly allocated to planting positions. In the experiment, the trees are arranged according to the 'broken-stick' design outlined by Bruelheide et al.[49]. This design involves dividing the species pool into two equal groups for each subordinate richness level. From the total pool of 24 species, we worked with eight tree species: *Castanea henryi* Rehder & E.H. Wilson, *Castanopsis sclerophylla* (Lindl. & Paxton) Schottky, *Choerospondias axillaris* (Roxb.) B.L.Burtt & A.W.Hill, *Liquidambar formosana* Hance, *Nyssa sinensis* Oliv., *Quercus serrata* Murray, *Sapindus mukorossi* Gaertn. and *Triadica sebifera* (L.) Small (see Supplementary Table 3 for details on species family); in plots ranging from the monoculture to the 8-species mixture passing through 2- and 4-species mixtures. Hence, all species are equally represented at every species richness level.

### Field sampling

Sampling of leaves used for the main analyses (hereafter referred to as regular set) took place from mid-August to mid-September 2023. In every plot, we randomly chose six individuals from every species, and every species was sampled in two plots at each diversity level (see Supplementary Figs. 17, 18 to see the spatial arrangement of the sampled trees within each plot and how pairwise distance among tree remained constant along the diversity gradient). This results in a total of 384 trees from eight different species in 30 plots. However, the theoretical number of 384 trees was reduced to 381 due to the high mortality of *Triadica sebifera* in one of the plots, where we found only three individuals for that population. In order to capture the variability of the whole individual, from each tree we collected 12 fully developed leaves free from apparent mechanical or pathogen damage at three different heights and four different orientations of the crown, resulting in 4572 leaves. Immediately after collection, leaves were stored in sealable plastic bags with moistened tissue. Samples were transported in an isothermal bag equipped with cooling bags to prevent dehydration. In the laboratory, samples were temporarily stored at 6–8 °C for a maximum of 12 h before further processing.

In addition, we collected a set of leaf samples that was independent from the regular set in order to predict the leaf economics spectrum (LES) trait values for the samples of the regular set based on the relationship between reflectance spectra and measured trait values of the calibration set (Supplementary Fig. 19; see leaf trait prediction section for details). For the calibration set, we included 20 leaf samples per species across all species richness levels, collected at different heights and orientations within the crown, in order to maximize the variability of trait samples for each species (i.e., combinations of species considering closest neighbors, different positions of the leaf within the crown, and the tree's location within the experiment). Each of the 160 samples was composed of four leaves on average depending on the leaf size, to ensure sufficient material for laboratory analyses.

### Spectroscopy and laboratory analyses

Visible-near infrared spectrometry (Vis-NIRS) is emerging as a high-throughput phenotyping technique to manage large sample sizes and predict individual leaf trait values using calibration models[41,74]. For all leaves (regular and calibration samples), we acquired reflectance spectra with a portable Vis-NIRS device (ASD FieldSpec4 Wide-Res Field Spectroradiometer, Malvern Panalytical Ltd, Almelo, Netherlands). Reflectance was measured across the full range of the solar radiation spectrum (350-2500 nm) by taking three repeated measures on the adaxial side of each leaf while avoiding main veins. The equipment was optimized regularly with a calibration white panel (Spectralon, Labsphere, Durham, New Hampshire, USA). For each measurement, ten spectra were averaged internally to reduce noise. A splice correction was applied to the spectral data to minimize the disjunctions between the three sensors of the ASD FieldSpec (VNIR, SWIR1 and SWIR2, with ranges 350-1000 nm, 1001-1800 nm and 1801-2500 nm, respectively). Therefore, the splicing regions were configured according to the points between sensors (from 750 to 1000 and from 1800 to 1950[75]). After splice correction, outlier detection was performed by using a similar procedure as in Li et al.[64]. First, all spectra were visually inspected in the laboratory after acquisition. Additionally, for every species separately, we calculated the local outlier factor of every spectrum[76] and hence considered as outliers 25 spectra that had a value higher than 2 for the local outlier factor (Supplementary Fig. 20). Finally, we excluded the spectral region between 350 and 399 nm for subsequent analyses due to the typical large amount of sensor noise in this region[77].

For the samples of the calibration set, we determined five morphological and chemical leaf functional traits which are assumed to reflect a plant's strategy in terms of the investment of nutrients and dry mass in the leaves[60,78] and are key components of the leaf economics spectrum[58,59] (Supplementary Fig. 21): specific leaf area (SLA; leaf area divided by leaf dry mass; mm²/mg), leaf dry matter content (LDMC; leaf dry mass divided leaf fresh mass; mg/g), carbon content (C; %), nitrogen content (N; %), and phosphorus content (P; μg/g). After collection, the saturated fresh leaves of the calibration LES samples were weighed (DeltaRange Precision Balance PB303-S; Mettler-Toledo GmbH, Gießen, Germany) and scanned at a resolution of 300 dpi to measure leaf area (WinFOLIA; Regent Instruments, Quebec, QC, Canada). Leaves were oven-dried at 80 °C for 72 h and weighed to calculate SLA and LDMC. Dried leaves were ground (Mixer Mill 400; Retsch, Haan, Germany), and 200 mg of the resulting powder was used for a nitric acid digestion. After the digestion, P was measured through a molybdate spectrophotometric method (UV-VIS Spectrophotometer UV-1280; Shimadzu, Duisburg, Germany)[79]. Additionally, we used an elemental analyzer (Vario El Cube; Elementar, Langenselbold, Germany) to gas-chromatographically determine C and N contents.

### Leaf trait prediction

The calibration dataset (spectral data and corresponding trait measurements) was then divided into train and test sets, which account for a proportion of 75% and 25%, respectively. We used a convolutional neural network (CNN) approach for leaf trait prediction based on the spectral data[33,80]. First, input spectra from the train and test sets were augmented from 2501 to 12,255 wavelength features by using transformations based on a combination of standard normal variates and Savitzky-Golay derivatives[81]. Then, a CNN composed of one convolutional layer followed by three dense layers was fitted to train the samples. To avoid overfitting, batch normalization was applied after the convolutional layer[81]. Hyperparameter tuning for every CNN was performed independently for every trait, by adjusting the number of filters, their size for the convolutional layers, and the number of nodes in the dense layers (Supplementary Table 7). For model optimization, an Adam algorithm and a loss function based on the mean squared error was used. All species were used together in a CNN to provide

greater spectral and trait variability in our training and test sets. We tested the predictive ability of the CNNs by assessing the coefficient of determination ($R^2$) and the root mean squared error (RMSE) for the predicted and measured values in the test set and in the train set. The mean $R^2$ of the test set was 0.74 ± 0.15 (mean ± standard deviation), with a maximum $R^2$ for SLA and LDMC (both 0.91) and minimum for P (0.54). The mean $R^2$ of the train set was 0.80 ± 0.14, with a maximum for LDMC (0.96) and minimum for P (0.62; Supplementary Figs. 22, 23). These trained CNNs were used for predicting trait values of leaves from the regular set of samples. After leaf trait prediction, we excluded, on average across all traits, 3.46% of the predicted trait values (1.35% for SLA, 1.78% for LDMC, 3.31% for C, 3.85% for N and 5.14% for P; Supplementary Fig. 24, Supplementary Table 8) as they lay outside the interval formed by the median, plus or minus 3 median absolute deviations[82]. This threshold for excluding predicted data was chosen as these values seemed unrealistic and were negative in some cases (Supplementary Fig. 25). The suitability of our sample size for the use of CNN for predicting leaf traits was assessed by evaluating changes in the $R^2$ under different scenarios of completeness of the training set, thus, simulating the predictive ability of the CNNs when we only use a subset of samples from the training set (see Supplementary Fig. 26). Leaf trait predictions and consecutive statistical analyses were conducted in the R environment with R version 4.2.3[83].

## Metrics of intraindividual and intraspecific functional trait variability

We identified the main axes of functional trait variability by performing principal component analyses (PCA) on the scaled predicted functional traits of all our leaves. Then, by using a Horn's parallel analysis to choose axes of trait variability, as implemented in the paran package[84], we selected the first two axes, which accounted for 79.07% of the variability in our dataset (53.43% and 25.64% explained by the first and second axis, respectively; see Fig. 1, and Supplementary Fig. 3, 4, Supplementary Table 1) and showed adjusted eigenvalues > 1 (2.63 and 1.26 for the first and the second axis, respectively). Due to the presence of missing values in our dataset as a consequence of the removal of extreme predicted trait data (see Leaf trait prediction section), missing values were imputed using a PCA-based method as implemented in the missMDA package[85] for every species independently prior to the PCA described above. This procedure, while avoiding unrealistic values, may also underestimate intraspecific and intraindividual variation.

The selected axes were used to measure the leaf intraindividual and intraspecific trait variability of a given individual or population, respectively, by estimating trait probability densities[39,86] using the package TPD[51] (Supplementary Fig. 27). Therefore, we assessed trait variability by considering probabilistic multivariate trait distributions with two dimensions (PC1 and PC2). Further, trait variability for single axes (for PC1 and PC2 independently) was also assessed. First, by considering leaf-level values on PC1 and PC2, we compiled a trait probability density for every individual tree as an approach to intraindividual trait variability. We estimated the bandwidth of the kernel functions by using an unconstrained bandwidth matrix as implemented in the ks package[87] and applied a 5% quantile threshold to the trait probability densities. In order to calculate intraspecific trait variability, we first assessed the mean PCA scores of every tree individual by using a bootstrap approach[46] and used the individual-level data to assess trait probability densities for all populations (following the same procedure as described for the trait probability densities of individuals). In both cases (individual and species levels), from the trait probability densities, we calculated two functional indices that describe two components of trait variability: (1) functional richness (FRic) indicates the total extent of the trait probability density and aims to detect changes in the niche space of individual trees and of the populations[39,88], respectively, and (2) functional divergence (FDiv) indicates the degree to which the abundance within the functional trait

space is distributed toward the edge of the functional volume and, therefore, describes whether the distribution of leaves and tree individuals, respectively, in the trait space is clustered or dispersed[39,51,89]. Last, the trait probability densities measured for individual trees based on leaf-level data were used to assess the intraspecific trait overlap as the mean overlap between all the trees belonging to the same population[39]. Both functional indices and intraspecific trait overlap were estimated by using the TPD package[51].

## Metrics of intraindividual and intraspecific spectral variability

We used the leaf reflectance spectra collected for the regular set in order to study changes in spectral variability. Leaf reflectance spectra are integrative measures of the leaf phenotype and reflect morphological, physiological, and chemical characteristics related to the plant's growth strategy[63], making these measurements ecologically meaningful. However, the multidimensionality of this data results in complexity for management and analysis[90], making difficult the assessment of leaf spectral variability. Therefore, prior to analyses, we reduced the number of dimensions in the spectral data by identifying segments of the leaf reflectance spectrum that can be summarized by one unique principal component. To do so, we used a Hierarchical Spectral Clustering with Parallel analysis (HPS-CA), a data-driven dimension reduction approach originally developed for the segmentation of human facial features[91,92]. As in Li et al.[90], we first used a Horn's parallel analyses[84] on all the wavelengths for which reflectance was measured (each of the 2101 features measured between the 400 and 2500 nm) in order to see which principal components were retained, assuming that a principal component is retained when its associated eigenvalue is higher than 1 (see Supplementary Fig. 6a). When more than one component was retained, the wavelengths were clustered into two groups (i.e. segments) by using spectral clustering as implemented in the kernlab package[93]. This process was then repeated in each generated segment until a segment for which only one principal component was retained (supplementary Fig. 6b, c) was identified. We identified 56 segments of which 29 retained only one principal component (Supplementary Fig. 7). The principal components retained by these segments were then used as spectral traits to calculate the same functional indices described above (FRic and FDiv) at the intraindividual and intraspecific level as described above for leaf functional traits.

## Statistical analyses

To assess the effect of tree species richness on leaf intraindividual and intraspecific functional trait variability (for both multivariate functional indices and functional indices for PC1 and PC2) and spectral trait variability (for the 29 principal components identified using the HSC-PA), we used linear mixed-effects models (LMMs) with the functional indices as a response variable and tree species richness (log2-transformed) as a fixed factor. In addition, tree diameter at breast height (DBH) and slope of the terrain in the position of the tree were included as covariates in the model for intraindividual variability, while the mean slope of the terrain in the plot was included as a covariate in the model for intraspecific trait variability due to its variability across the study site (Supplementary Fig. 16). In a first step, aspect (measured as a categorical variable) was considered as a covariate in our model, but it was afterwards discarded due to the low importance of this variable (see Supplementary Fig. 28, and Supplementary Table 9). We included species identity and plot identity nested in tree composition of the plot as crossed random effects in the models of intraindividual trait and spectral variability, and species identity and tree composition of the plot as random effects in the models of intraspecific trait and spectral variability. We used diagnostic plots of the residuals to study the assumptions of normality, homoscedasticity and linearity in our models: residuals vs fitted values plots, histograms of the residuals and Q-Q plots for the deviance of the residuals. Then, we tested the

significance of fixed effects using a two-sided likelihood ratio tests[94] and assessed standard estimates (β) as effect sizes using the effectsize package[95]. Finally, we assessed the quality of fit of our model by calculating the marginal and conditional $R^2$, which address the variance explained only by fixed effects and the variance explained by the entire model including the random effects, respectively.

In order to assess the effects of tree species richness on intraspecific leaf trait overlap and how this effect is mediated by the intraspecific and intraindividual leaf functional trait variability, we used a Piecewise Structural Equation Model (piecewise SEM) as implemented in the piecewiseSEM package[96]. Here, species identity was included as a random effect. First, we defined the conceptual model as a set of regressions, representing the relationships between the variables and fit linear mixed models (LMMs) based on these relationships (Supplementary Fig. 8). Correlated error terms were included between indices of intraindividual functional trait variability and between indices of intraspecific functional trait variability. The mean slope of the plot was included as a covariate in the models for intraspecific and intraindividual functional trait variability. Then, the model fit was evaluated based on d-separation test and Fisher's C statistics[96]. Eventually, as intraindividual and intraspecific functional divergence did not show any significance and weak standard estimates, we reduced the SEM by excluding these two metrics and the correlated error terms. All of these results remained qualitatively similar when using the full and the reduced SEM (Supplementary Fig. 10).

## Null models for functional diversity

In order to assess the effects of intraspecific and intraindividual functional trait variability in the assessment of functional diversity of a community and its relationship with species richness, we used null models that randomized different sources of trait variation occurring within the species.

First, observed functional diversity in every plot was assessed by using sums of trait probability densities from the leaf-level to the community level, therefore, expanding to the individual level the trait probability density framework for functional diversity developed in Carmona et al.[39] (Supplementary Fig. 11). Thus, based on the leaf-level data (level 1), we estimated trait probability density for individuals (level 2) and, afterwards these trait probability densities were summed at the species level (considering the given species in a population; level 3). Finally, by summing the trait probability densities of the different populations occurring in a community we obtained final trait probability densities at the community level (level 4). The community trait probability densities were then used to assess FRic and FDiv as metrics of functional diversity in a plot. For this last step, the contribution of the trait probability density of every population was weighted according to the sum of wood volume of every species in the central area of every plot (including the 36 trees in the center of the plot). To assess the wood volume (WV) of the trees, basal area and height were measured in 2022 and the conversion factor calculated by Huang et al.[97] for our study species in our study site was used to estimate wood volume as:

$$WV = 0.5412\,m^3 m^{-3} - 0.1985\,m^{-3} \times basal\ area \times height \quad (1)$$

Following this framework, we ran simulations randomizing hierarchically different sources of variation occurring within the species (Supplementary Fig. 12). Therefore, these null models simulated communities with the same species composition and abundances, but they randomized data on different steps of the framework for measuring functional diversity:

(1) Random population null model: The trait probability densities of every population were calculated based on observed data and,

afterwards, these population trait probability densities were shuffled for every species. This model aims to test which is the effect of considering the functional identity of the population in the plot.

(2) Random tree null model: The trait probability densities of all trees were calculated based on the observed leaf values and the trees were shuffled for every species. This model aims to test the effect of intraspecific variability on community functional diversity.

(3) Random leaf null model: Leaf-level functional trait values were shuffled for every species before calculating functional diversity. This model aims to detect the whole effect of the variability occurring within species (intraspecifically and intraindividually) on functional diversity.

As the intraindividual variability tends to be clustered around the centroid of every tree and trees in the same population are more similar to each other compared to trees from other populations (Supplementary Fig. 4), the random leaf null model could represent highly unrealistic scenarios. Therefore, we decided to build another null model for the assignment of random leaves in which the pool of leaves was more restricted than in the random leaf null model:

(4) Population-restricted random leaf null model: Leaf-level functional trait values were shuffled for every population before calculating functional diversity. This model aims to detect the effect of the whole variability occurring within populations (intraspecific and intraindividual variability) in functional diversity.

We simulated 500 null assemblages for every plot and every type of null model. We visually inspected the changes and stabilization of the mean and variance of every null distribution with an additive number of simulations (Supplementary Fig. 29). Finally, to assess the differences between the observed and the simulated values of the functional indices we used standardized effect sizes (SES) as in Gotelli and McCabe[98]

$$SES = \frac{FD_{observed} - mean(FD_{simulated})}{SD(FD_{simulated})} \quad (2)$$

Where FD corresponds to any of the measured functional indices. SESs were calculated independently for every plot and type of null model. To test the effects of tree species richness and the type of null model on the SES for every functional index, we used LMMs and included the plot identity nested in tree species composition, as a random effect. In every model, we included the SES of every functional index as a response variable. Thus, we fitted two models with tree species richness (log2 transformed), type of the null model, and their interaction as response variables. Then, we tested the significance of fixed effects by using two-sided likelihood ratio tests, following the same procedure described previously for LMMs fitted for the intraspecific and intraindividual trait variability. All LMMs were fitted using the lmer function in the lmerTest package[99]. We considered that the fitted null model coefficients were significantly different from the random scenario when the 95%-confidence intervals did not overlap with zero.

## Reporting summary

Further information on research design is available in the Nature Portfolio Reporting Summary linked to this article.

## Data availability

All materials needed to evaluate the conclusions in the paper are present in the paper and/or the Supplementary Information. The data used in this study are available at the Zenodo repository[100] with the identifier (https://doi.org/10.5281/zenodo.14190699).

## Code availability

R codes used in this study are available at the Zenodo repository[100] with the identifier (https://doi.org/10.5281/zenodo.14190699).

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

## Acknowledgements
We are grateful to Leana Meder, Michael Köhler and Georg Hähn for the assistance in data collection, to Helge Bruelheide for the suggestions in data analysis and to Cheng Li for the support with the analyses on leaf spectral data. Further, we acknowledge the support of the BEF-China platform, the TreeDì coordinators, Stefan Trogisch and Xue Kai, and TreeDì members. This study was supported by the International Research Training Group TreeDì, jointly funded by the Deutsche Forschungsgemeinschaft (DFG, German Research Foundation) 319936945/GRK2324 and the University of Chinese Academy of Sciences (UCAS). C.P.C. was supported by the Estonian Research Council (PRG2142), the European Union (ERC, PLECTRUM, 101126117) and the Spanish Ministry of Science, Innovation and Universities through the ATRAE Program 2024 (ATR2024-154934).

## Author contributions
Conceptualization: P.C.S-B. and S.H., with support from C.P.C. and M.C.S. Methodology: P.C.S-B., with support from C.P.C., M.C.S., R.B. and S.H. Investigation: S.L. and X.L. collected the tree basal area and height data. P.C.S. collected the leaf trait data with support from L.S. and S.H. Data curation: P.C.S-B. Formal analysis: P.C.S-B., with support from C.P.C., M.C.S., R.B. and S.H. Visualization: P.C.S-B. Supervision: S.H. Writing—original draft: P.C.S-B. Writing—review & editing: P.C.S-B., C.P.C., M.C.S., R.B., L.S., X.L. and S.H.

## Funding

## Competing interests
The authors declare no competing interests.
