## [Transparent Peer Review file · Nature Communications]

Intraspecific and intraindividual trait variability decrease with tree richness in a subtropical tree biodiversity experiment

Corresponding Author: Mr Pablo Castro Sánchez-Bermejo

Version 0:

Reviewer comments:

Reviewer #1

(Remarks to the Author)

I had the opportunity to revise the MS NCOMMS-24-84869 submitted by Pablo Castro Sánchez-Bermejo and colleagues. The MS is focused on a hot topic of functional ecology and presents interesting results on a challenging and time-consuming experiment with noticeable field data collection.

The authors reported that intraspecific and intraindividual trait variability significantly decrease with higher tree species richness in a tree biodiversity experiment. The results are well supported by the analyses, the discussion is clear, and the figures are extraordinarily understandable. Regarding the introduction, despite it is well written, I found an abrupt change in the first part (from LL76-78) where the topic shifts from "species" to "trees". So, the authors abruptly change their approach from general expectations to a particular growth form. This is a bit confusing. If they want to introduce the topic only for trees, it must be clear from the beginning. If trees are considered only a subset of plants in which these theories can be easily tested, it must be stated.

Regarding the method, probably more information are needed:

L494. Plot dimensions?

L509. Did you register the "coordinates" of the collected trees? The spatial arrangement and the distance between individuals from which traits have been collected may affect the results, considering that your topic is on "interactions".

L518. Independent sets? What do you mean?

(Remarks on code availability)

Reviewer #2

(Remarks to the Author)

This study addresses an intriguing question: how intraspecific trait variation (differences among individuals of the same species) and within-individual trait variation (plasticity within a single individual) respond to tree species richness. The authors further explore the relative contributions of these two sources of variation to community-level functional diversity. While the topic is compelling and ecologically significant, the execution raises several concerns. Specifically, the theoretical framework lacks clarity in justifying the hypotheses, and key methodological approaches appear insufficiently rigorous, potentially introducing significant biases that undermine the reliability of the conclusions. My primary concerns are as follows:

1. Lines 72-75: The authors' hypothesis that intraspecific trait variability is greater in species-poor communities—attributed primarily to intraspecific competition—requires stronger theoretical justification. While intraspecific competition may intensify in low-diversity communities, this reasoning oversimplifies a complex ecological dynamic. For instance, interspecific competition, which is often stronger in species-rich communities, could also drive trait variation. Furthermore, the relationship between competition intensity and intraspecific trait variability remains ambiguous: high competition (whether intra- or interspecific) could theoretically increase trait variability by favoring resource partitioning, but competition intensity itself does not necessarily scale linearly with species richness. Without addressing these interacting mechanisms, the authors' expectation of higher intraspecific trait variability in low-diversity communities appears inadequately supported. Similar issues arise in the rationale linking within-individual trait variation to species richness (lines 75-78).

2. Figure 1: While the authors present an elegant conceptual illustration of expected trait density patterns across a species richness gradient, their assumption that intraspecific trait variability remains lower in high-diversity communities requires further scrutiny. The logic appears inconsistent with ecological theory: if species adapt their traits to heterogeneous abiotic conditions (e.g., light, soil gradients), individuals within a species may exhibit multimodal trait distributions. In such cases, intraspecific trait variability could remain high even in species-rich communities, contradicting the authors' generalized expectation. This raises a critical limitation: without empirical trait distribution data for each species and trait, it is impossible to validate whether unimodal vs. multimodal patterns align with their assumptions. Until such empirical validation is provided, the proposed relationship between species richness and intraspecific trait variability risks being oversimplified.

3. A critical limitation arises from the authors' sampling design and model reliability. The study relies on only 20 leaf samples per species across all species richness levels (Line 524), with the majority of trait values (4,568 leaves) derived from model predictions rather than direct measurements (Lines 580–600). This approach raises two interrelated issues. First, the models used to predict trait values demonstrate limited predictive accuracy, as evidenced by their low explanatory power on test data (Lines 595, 597). Second, the small empirical sample size (20 leaves/species) severely constrains the models' ability to capture the true range of trait variation, particularly multimodality. Without sufficient independent empirical data, it is impossible to robustly evaluate whether observed or predicted trait distributions reflect biological reality (e.g., multimodal patterns driven by environmental adaptation) or are artifacts of undersampled data and weak model performance.

4. Moreover, the study's reliance on a single global performance metric (R^2) to evaluate model accuracy obscures critical species-level variations in predictive reliability. For convolutional neural network (CNN) models, which are inherently sensitive to input data distribution, reporting species-specific performance metrics—such as confusion matrices or per-species error rates—is essential to assess potential biases. The authors' omission of such granular analyses raises concerns, as model accuracy likely varies across species due to differences in trait complexity, or environmental representation. This differential bias, in turn, risks distorting patterns of intraspecific trait variability across the species richness gradient and invalidating comparisons between communities. The authors do not address whether such biases could amplify or obscure the hypothesized relationships between diversity and trait variation, leaving the robustness of their conclusions in question.

5. A defining feature of the BEF-China experiment, distinguishing it from other BEF studies, is its incorporation of substantial environmental heterogeneity. Such heterogeneity, particularly in topography and microsite conditions, is well-documented to drive intraspecific trait variation by influencing individual-level trait expression. While the authors account for slope as an abiotic factor in their model (Line 651), this represents only a partial acknowledgment of the experiment's environmental complexity. For instance, terrain aspect (e.g., north- vs. south-facing slopes) is a critical yet unaddressed variable that directly affects light availability, moisture regimes, and thermal conditions, all of which can induce significant trait plasticity within species. The exclusion of such factors raises concerns about whether the models adequately capture the environmental drivers of trait variation. If unmeasured variables systematically influence trait expression, the study's conclusions about intraspecific variability and its relationship to species richness risk being confounded by residual environmental noise.

(Remarks on code availability)

Version 1:

Reviewer comments:

Reviewer #1

(Remarks to the Author)

The authors correctly implemented the reviewers' comments. In my opinion the paper can be accepted.

(Remarks on code availability)

Reviewer #2

(Remarks to the Author)

Thank you very much for your detailed and thoughtful responses to my earlier comments and questions. While some points have been clarified satisfactorily, several important issues remain unresolved. I would appreciate your further consideration of the following concerns:

1. The introduction currently devotes substantial discussion to how intraspecific trait variability changes with species richness (e.g., the second paragraph). However, your core result (e.g., Fig. 2) primarily addresses how community-level functional diversity changes with richness. There is minimal theoretical framing in the introduction regarding the expectations for results like Fig. 2. This disconnect makes it difficult to interpret the results and understand their theoretical contribution upon reaching them. Crucially, which main figure or table in the results directly addresses the unresolved theoretical questions raised in the second paragraph of the introduction? The current structure creates a sense of

disconnection between the introduction and the results.

2.Regarding my previous concern about potential sample size limitations (Point #3), your explanations focused on your sampling design philosophy and model robustness. While informative, these do not empirically demonstrate that 20 leaves per species are sufficient to capture the full range of intraspecific trait variation, especially potential multimodality. A more direct approach to address this would involve additional sampling. For instance, selecting a few representative species and measuring a larger number of leaves (e.g., 100 per species) would allow a quantitative assessment of how well the original 20-leaf samples estimate the true intraspecific trait variation within those species.

3.The supplementary Table 5 you provided is very helpful and clearly shows that model performance (R^2) is low for many species-trait combinations. You cited two potential reasons: (1) the "smaller calibration set" and (2) the "inherently weaker spectral signal associated with stomatal anatomical traits". Point 1: Acknowledging the small calibration set size implicitly suggests it might be insufficient. Would it not be more advisable to conduct additional measurements to strengthen the calibration for these traits? Point 2: If the spectral signal for stomatal traits is inherently weak, their inclusion in analyses based on predicted values seems problematic, potentially introducing significant noise. A more reasonable approach would be to exclude these traits when calculating community-level functional diversity indices derived from spectral predictions. While I appreciate the ecological importance of stomatal traits ("the importance of these traits in leaf strategies, reflecting a gradient mostly related to water use, and because they have been shown to be relevant in tree-tree interactions"), if they are crucial, the solution should be to directly measure them for the relevant samples and do the functional diversity calculations on these empirical values, rather than relying on potentially unreliable predictions from spectral data.

4.I appreciate the inclusion of CV analysis of spectral data (Supplementary Figs. 13 & 14) as supporting evidence. However, the figures show considerable variation. To convincingly demonstrate that the spectral results align with the main trait-based findings (i.e., lower variability at low richness), formal statistical testing of the relationship between species richness and spectral CV is necessary. Furthermore, to fully leverage this alternative approach, it would be highly informative to present the core results (analogous to your main Fig. 2 and 3) derived solely from the spectral data within the main manuscript.

5.At last, thank you for addressing the aspect of orientation and confirming its lack of systematic influence. However, an additional concern arises: is there a systematic difference in the CV of environmental factors across plots of different species richness levels? For example, what pattern emerges if the y-axis in a figure like Fig. 2 represented the CV of key abiotic factors (e.g., soil moisture, light availability) instead of trait diversity? Heterogeneity in abiotic environmental conditions is a known driver of both intraspecific trait variation and functional diversity, and its potential correlation with the manipulated species richness gradient needs examination.

(Remarks on code availability)

Version 2:

Reviewer comments:

Reviewer #2

(Remarks to the Author)

I have carefully read the author's revisions and responses, and all of my previous questions have been thoroughly addressed. I would like to thank the author for their significant effort in strengthening this work. I have no further comments.

(Remarks on code availability)

We thank the reviewers for the detailed evaluation that has helped to improve the manuscript, as well as to clarify the parts which were not explained sufficiently. We address all questions and comments from the reviews in detail below, and provide a line-by-line account of the changes made in the substantially revised manuscript. The comments provided by the reviewers led to important changes in the introduction and presentation of the hypotheses and helped us to clarify part of the methods, especially regarding the use of leaf spectroscopy to predict leaf functional traits. However, these revisions did not result in major changes to the results or the main conclusions of the study. In addition, minor changes were made in the manuscript in order to fit the guidelines of the journal and improve the readability of the draft. Please note that the line numbers refer to the clean version without track-changes.

Reviewer #1 (Remarks to the Author):

I had the opportunity to revise the MS NCOMMS-24-84869 submitted by Pablo Castro Sánchez-Bermejo and colleagues. The MS is focused on a hot topic of functional ecology and presents interesting results on a challenging and time-consuming experiment with noticeable field data collection.

The authors reported that intraspecific and intraindividual trait variability significantly decrease with higher tree species richness in a tree biodiversity experiment. The results are well supported by the analyses, the discussion is clear, and the figures are extraordinarily understandable.

>> We thank reviewer #1 for the interest in the manuscript, as well as the supportive comments regarding the analyses, the discussion and the figures.

Regarding the introduction, despite it is well written, I found an abrupt change in the first part (from LL76-78) where the topic shifts from "species" to "trees". So, the authors abruptly change their approach from general expectations to a particular growth form. This is a bit confusing. If they want to introduce the topic only for trees, it must be clear from the beginning. If trees are considered only a subset of plants in which these theories can be easily tested, it must be stated.

>> We agree that this distinction was not clear enough. We made several changes to the manuscript to account for this and to make clear that our assumptions apply to plants in

general, but that we use trees as model organisms to test our hypotheses because the topic is most relevant for this particular growth form. The use of trees is justified as these are long-living organisms with a complex architecture that allows them to express substantial variability within and among individuals (typically via plasticity)¹. We now refer mainly to plants in the more general part of the introduction but note that examples given in lines L83-87, L97-100 and L124-127 are focused on trees due to the availability of literature and the relevance of the studies mentioned to discuss the results later. We also refer to the use of trees as model organisms in our study in the last paragraph (L151-154), prior to describing the main objectives of the study.

Regarding the method, probably more information are needed:

L494. Plot dimensions?

>> Each plot consisted of 400 trees planted in 20 rows and 20 columns. Planting distance is 1.29 m in every case. As a result, the dimension of each plot is 25.8 by 25.8 m. As this was not clear in the main text, we added this information in L488.

L509. Did you register the "coordinates" of the collected trees? The spatial arrangement and the distance between individuals from which traits have been collected may affect the results, considering that your topic is on "interactions".

>> Following the reviewer's concern, we included a new figure in the Supplementary Information (Supplementary Fig. 16) that illustrates the spatial arrangement of each sampled tree within the plots where it was sampled. The selection of the sampled trees was, in every case, random for each species in each plot (except for *Triadica sebifera* in T9, for which the mortality of the species in the plot was high and, therefore, all remaining individuals were sampled). Mean distances between conspecific trees remained constant along the diversity gradient (Supplementary Fig. 17a). In addition, R^2 between distances and tree richness (log₂-transformed) was 0.006, thus reflecting a lack of correlation between these two variables (non-significant using a linear mixed-effects model having distance of conspecifics as a

response, tree richness (log₂-transformed) as a fixed factor and plot and species identity as crossed random effects).

Additionally, we tested how distances between conspecific trees differed between plots and for each species independently. We used linear models with distances as a response and plot as a fixed factor. Afterwards, we used post-hoc pairwise t-tests to study differences among plots. The results showed significant differences restricted to a few plots in the case of *Castanea henryi*, *Liquidambar formosana*, *Nyssa sinensis*, *Quercus serrata* and *Sapindus mukorrosi* (Supplementary Fig. 17b, e, f, g, h).

Since overall, there were no systematic differences in conspecific distances related to tree species richness, we stayed with our original analyses. To ensure transparency, we also included the spatial distribution of trees within plots in Supplementary Fig. 16 for readers' reference as we all the distribution of mean pairwise distances along the diversity gradient in Supplementary Fig. 17.

Supplementary Fig. 16. Spatial arrangement of sampled trees within each sampled plot. In each plot, 400 saplings were planted in 2009 following a grid pattern. Closest neighbors were planted at a constant distance of 1.29 m, resulting in plots of 25.8 by 25.8 m. Each planting position is represented by a circle for each plot, with grey circles indicating the position of non-sampled trees, while colored circles indicate the position of sampled trees. Different species are represented in different colors.

Supplementary Fig. 17. Differences in spatial distances between trees along the diversity gradient and among plots. We evaluated pairwise distances between conspecific trees along the tree species richness gradient by using a linear mixed-effects model and (a) results revealed no significant correlation ($P = 0.46$; $N = 948$). The linear mixed-effects model included tree richness (log₂-transformed) as a predictor and plot identity and species identity as crossed random effects. We evaluated differences in pairwise distances among plots for each species independently (b-i) and results showed specific differences in pairwise distances for some specific plots in the case of (b) *Castanea henryi*, (e) *Liquidambar formosana*, (f) *Nyssa sinensis*, (g) *Quercus serrata* and (h) *Sapindus mukorossi*. (post hoc t-tests).

L518. Independent sets? What do you mean?

>> In order to predict leaf functional traits from spectral data we needed to collect additional leaf samples in which we measured leaf spectral reflectance, but also functional traits (referred to as calibration set in the main text). However, as the method we used for measuring traits from the leaf economics spectrum (LES) was not compatible with the method we used for taking leaf imprints for stomatal traits, we collected two different sets. This was because the use of stomata imprints would leave nail polish traces which can compromise the reliability in the measurements of LES traits. At the same time, the method for measuring LES traits was destructive (requiring drying, weighing, and grinding of leaf samples), and therefore not compatible with the method for measuring stomatal traits, which required intact leaf surfaces for microscopy. Therefore, this resulted in the following sampling sets:

-Regular set: Leaves for which only leaf reflectance was measured and that were used for the prediction of leaf functional traits used for the main analyses.

-Calibration LES set: Leaves for which we measured leaf reflectance and five LES traits: specific leaf area (SLA), leaf dry matter content (LDMC), leaf carbon content (C), leaf nitrogen content (N) and leaf phosphorus content (P). These samples were used to train the convolutional neural networks that predicted LES traits from spectral data.

-Calibration stomata set: Leaves for which we measured leaf reflectance and two leaf functional traits related to stomata morphology: stomatal density (SDens) and stomatal size (SSize). These samples were used to train the convolutional neural networks that predicted stomatal traits from spectral data.

As this may have been unclear in the main text, we included changes in L500 when referring to the regular set and in L513 when describing the calibration sets. Additionally, we included a new figure (Supplementary Fig. 18), which illustrates the workflow and use of the three different sample sets collected.

Supplementary Fig. 18. Analytical workflow used to generate a trait matrix from spectral data and two calibration sets. First, (a) we used a calibration leaf economics spectrum (LES) set and a calibration stomata set of leaves to train convolutional neural networks that can predict LES traits and stomatal traits, respectively, from spectral data. The calibration sets are composed of samples of the eight tree species included in this study. For each sample in these sets, leaf reflectance in the range of solar radiation (350-2500 nm) was measured. Additionally, for the samples of the calibration LES set, five functional traits belonging to the LES were measured: specific leaf area (SLA), leaf dry matter content (LDMC), leaf carbon content (C), leaf nitrogen content (N) and leaf phosphorus content (P). For the samples of the calibration stomata set, two stomatal traits were measured: stomatal density (SDens) and stomatal size (SSize). In a second step, (b) trained convolutional neural networks were used to predict leaf functional traits from spectral samples of the regular set. The regular set comprises leaf-level spectral samples of eight tree species collected along an experimental species richness gradient with monocultures and mixtures of 2, 4 and 8 tree species.

#####

Reviewer #2 (Remarks to the Author):

This study addresses an intriguing question: how intraspecific trait variation (differences among individuals of the same species) and within-individual trait variation (plasticity within a single individual) respond to tree species richness. The authors further explore the relative contributions of these two sources of variation to community-level functional diversity. While the topic is compelling and ecologically significant, the execution raises several concerns. Specifically, the theoretical framework lacks clarity in justifying the hypotheses, and key methodological approaches appear insufficiently rigorous, potentially introducing significant biases that undermine the reliability of the conclusions.

>> We thank reviewer #2 for the helpful feedback. Comments raised by the reviewer regarding the framework, hypotheses and methods are addressed below.

My primary concerns are as follows:

1. Lines 72-75: The authors' hypothesis that intraspecific trait variability is greater in species-poor communities—attributed primarily to intraspecific competition—requires stronger theoretical justification. While intraspecific competition may intensify in low-diversity communities, this reasoning oversimplifies a complex ecological dynamic. For instance, interspecific competition, which is often stronger in species-rich communities, could also drive trait variation. Furthermore, the relationship between competition intensity and intraspecific trait variability remains ambiguous: high competition (whether intra- or interspecific) could theoretically increase trait variability by favoring resource partitioning, but competition intensity itself does not necessarily scale linearly with species richness. Without addressing these interacting mechanisms, the authors' expectation of higher intraspecific trait variability in low-diversity communities appears inadequately supported. Similar issues arise in the rationale linking within-individual trait variation to species richness (lines 75-78).

>> We thank the reviewer for the comment and agree that the relationship between trait variation and competitive interactions is intricate. Following this concern, we restructured the introduction and hypotheses of the study. Specifically, we now build our framework considering results from previous literature. Specifically, some studies concluded that intraspecific trait variability decreased with species richness while other studies found the

opposite result^{2,3}. Therefore, we discuss that these contrasting patterns in the direction of change of intraspecific trait variability may result from the following mechanisms:

- 1) Supported by the limiting similarity theory, which suggests that individuals can coexist only if they acquire resources differently, we would expect individuals from the same species to specialize and occupy different parts of their shared niche, resulting in greater variation among conspecifics, *i.e.*, diversifying selection. This is a kind of “niche sub-partitioning”. In this case, an increase in species richness results in a decrease in the probability of intraspecific interactions and so should dampen or reverse the effects of conspecific competition.
- 2) Hierarchy in plant-plant interactions, which considers that there is a competitively dominant phenotype, may act as a pressure on individuals to change trait expression. In this case, an increase in species richness results in an increase in the diversity of neighborhoods and competitive interactions within the same plot. This heterogeneity within the same community would eventually boost intraspecific trait variability.

We made important changes in the second paragraph of the introduction (L73-103) to consider this and to give more emphasis to the part that was already included in the previous version, stating that “these contrasting results among studies propose that there is no general direction of change of intraspecific trait variability in response to species diversity, but that it likely depends on the specific interaction partners” (L101-103), which supports the idea that different kinds of interactions could result in different patterns.

Regarding the paragraph on intraindividual trait variability, the reviewer suggested that a similar problem may be present. In this case, we also decided to build the rationale based on previous literature, which is scarce but consistently points out to a decrease in intraindividual trait variability with species richness^{4,5}. Therefore, apart from the limiting similarity rationale, we suggested that environmental factors related to the structure and spatial arrangement could explain the patterns of intraindividual variability (L120-124). That is why, here we propose that: (1) intraindividual trait variability is high in monocultures because it acts as a source of niche differentiation in intraspecific interactions (as proposed by Pross et al.⁴) and (2) high intraindividual trait variability may help to cope with stresses (e.g. abiotic stresses⁶) and variability in climate⁷ that, as we know in the case of forest stands, tend to be higher in monocultures compared to mixtures⁸.

Considering all this, we now consider two alternative hypotheses: the limiting similarity hypothesis, that assumes that dissimilarity in resource uptake is key to mediate tree-tree interactions, and the community heterogeneity hypothesis, that assumes that the presence of more diverse neighborhoods within the same plot increases the number of different niches for the same species. However, to fit the format of the journal, which specify that the introduction needs to conclude with a brief presentation of our main findings, the hypothesis section at the end of the introduction was removed and replaced by some main insights into the main results (L169-176). Nevertheless, these changes regarding the hypotheses were kept as part of the caption of Supplementary Fig. 1 (previous Fig. 1).

2. Figure 1: While the authors present an elegant conceptual illustration of expected trait density patterns across a species richness gradient, their assumption that intraspecific trait variability remains lower in high-diversity communities requires further scrutiny. The logic appears inconsistent with ecological theory: if species adapt their traits to heterogeneous abiotic conditions (e.g., light, soil gradients), individuals within a species may exhibit multimodal trait distributions. In such cases, intraspecific trait variability could remain high even in species-rich communities, contradicting the authors' generalized expectation. This raises a critical limitation: without empirical trait distribution data for each species and trait, it is impossible to validate whether unimodal vs. multimodal patterns align with their assumptions. Until such empirical validation is provided, the proposed relationship between species richness and intraspecific trait variability risks being oversimplified.

>> Following the reviewer's concern and also the changes in the framework mentioned before, we made important changes in Figure 1 (now Supplementary Fig. 1; see last paragraph for the reason to move it to the Supplementary Information). We described the expected results based on the two alternative scenarios proposed before (the limiting similarity hypothesis and the community heterogeneity hypothesis). The first one remains equal to the figure in the previous version, but the expected results for the second hypothesis (Supplementary Fig. 1c, d) consider the possible multi-modularity of the species in the heterogeneous environments. Therefore, these changes in the figure are aligned with the changes in the main text, which considered several aspects of the ecological theory to try to illustrate the different scenarios expected.

Further, this figure is intended primarily as a conceptual tool, rather than as an empirical representation. It illustrates plausible scenarios (based on existing literature) of

change of intraspecific and intraindividual trait variability and their effect on functional diversity expected for our study.

Last, the decision to move this figure into the Supplementary Information was motivated by the need to fit the format of the journal. Since the introduction needs to conclude with a brief presentation of our main findings, placing a figure with hypothesized patterns in close proximity risked confusing the reader. As a result, we decided to refer to this figure in the main text (L154), but leave it as part of the Supplementary Information so the reader can see what the main hypotheses of the work were.

Supplementary Fig. 1. Expected patterns along the experimental tree species richness gradient for two alternative hypotheses. In the case of the limiting similarity hypothesis, with increasing tree species richness, we expect (a) a reduction of intraspecific trait variability (i.e. differences between the mean trait values of individual trees) and of intraindividual trait variability (i.e. differences between the trait values within a tree; represented as error bars around points of individual mean trait values), which would result in increasing intraspecific overlap (i.e. shared trait space between trees belonging to the same population), while (c) an increase in intraspecific trait variability (but still a decrease in intraindividual trait variability as supported by previous literature^{16, 32}) is expected to occur according to the community heterogeneity hypothesis. Curves represent the trait distributions of populations, with inner stacked curves belonging to the trait distribution of tree individuals. The structure of trait variation within species can influence community functional diversity, and (b) for the limiting similarity hypothesis we expect functional diversity in observed communities (represented as a baseline with a grey dashed line) to be higher compared to the functional diversity of virtual assemblages (colored values) for which different sources of trait variation are randomized. Specifically, we expect that the functional diversity of observed communities would be more similar to those assessed with models that randomize the identity of the populations (pink points) compared to those assessed with models that randomize the identity of the trees (purple points), and the total pool of leaves within a species (blue points), respectively. In addition, we expect these differences to be higher with low tree species richness due to the importance of intraspecific and intraindividual trait variability in the functional diversity of species-poor communities. In contrast, (d) for the community heterogeneity hypothesis we expect an increase in the deviations from the null model with tree species richness, and that the observed functional diversity will be especially dissimilar to those assessed with models that randomize the identity of the trees (purple points).

3. A critical limitation arises from the authors' sampling design and model reliability. The study relies on only 20 leaf samples per species across all species richness levels (Line 524), with the majority of trait values (4,568 leaves) derived from model predictions rather than direct measurements (Lines 580–600). This approach raises two interrelated issues. First, the models used to predict trait values demonstrate limited predictive accuracy, as evidenced by their low explanatory power on test data (Lines 595, 597). Second, the small empirical sample size (20 leaves/species) severely constrains the models' ability to capture the true range of trait variation, particularly multimodality. Without sufficient independent empirical data, it is impossible to robustly evaluate whether observed or predicted trait distributions reflect biological reality (e.g., multimodal patterns driven by environmental adaptation) or are artifacts of undersampled data and weak model performance.

>> The reviewer raises a valid concern that the calibration sample size per species (20 leaves) may not be sufficient to capture the full extent of intraspecific trait variability, particularly considering the potential multimodal nature of these distributions. However, our approach still captures more trait diversity than if we would have limited our sample number to what we could assess by direct trait measurements within a feasible time frame and resources.

In our study we aimed at collecting few samples per species, but accounting for representing all species. As shown by Ji et al.⁹, greater spectral and trait diversity can boost the transferability of predictive models. Moreover, while it is true that when using predictive models across functional groups, the predictions within groups tend to be weaker than the comparisons across groups¹⁰, this does not seem to be the case when comparing species with similar leaf functional traits¹¹. In our case, rather than being completely different species, our species are rather similar in the range of their functional traits (see Supplementary Fig. 20) and their leaf spectral properties (see Supplementary Fig. 19). That is why, even though some of our species tend to be more acquisitive (e.g. *Nyssa sinensis*) than others (e.g. *Castanopsis sclerophylla*), they nonetheless share overlapping regions in trait space. Second, the calibration sample set attempted to maximize the variability for each species by collecting samples from different plots, diversity levels and parts of the canopy. As a result, the collected samples per species aimed at maximizing the variability of trait values included per species and do in fact seem to capture multimodal distributions. This had been emphasized in L518-524.

As a result, we advocate that the use of multi-species leaf trait predictive models in our case is a strength that can allow us to detect the multimodality of tree species. We now

provide a clearer explanation about the selection of a multi-species predictive model in lines L586, 587 as well as show the distribution of functional traits in Supplementary Fig. 20.

Supplementary Fig. 20. Distribution of trait values of the leaf economics spectrum and stomata calibration sets for each species. Density plots are shown for (a) specific leaf area (SLA), (b) leaf dry matter content (LDMC), (c) leaf carbon content (C), (d) leaf nitrogen content, (e) leaf phosphorus content (P), (f) stomatal density (SDens) and (g) stomatal size (SSize). Each row and color corresponds to a species. The trait values were measured in samples collected from the field (see 'Spectroscopy and laboratory analyses' section in the main text) and used to predict leaf functional traits from spectral data.

4. Moreover, the study's reliance on a single global performance metric (R^2) to evaluate model accuracy obscures critical species-level variations in predictive reliability. For convolutional neural network (CNN) models, which are inherently sensitive to input data distribution, reporting species-specific performance metrics—such as confusion matrices or per-species error rates—is essential to assess potential biases. The authors' omission of such granular analyses raises concerns, as model accuracy likely varies across species due to differences in trait complexity, or environmental representation. This differential bias, in turn, risks distorting patterns of intraspecific trait variability across the species richness gradient and invalidating comparisons between communities. The authors do not address whether such biases could amplify or obscure the hypothesized relationships between diversity and trait variation, leaving the robustness of their conclusions in question.

>> We now report the coefficient of determination (R^2) and root mean squared error (RMSE) of the models for the samples collected for each species individually in Supplementary Table 5. These results show that, while for most of the traits in the leaf economics spectrum, the predictive ability of the global model within groups is high (e.g. SLA, N); this predictive ability decreases for stomatal traits: stomatal density (SDens) and stomatal size (SSize). For instance, despite the large range of SDens (ranging from 108.98 mm^{-2} to 899.13 mm^{-2}), some of the RMSE values for were still high, indicating a compromise in the accuracy of the predictive model. This lower predictive ability likely reflects a combination of factors, including the smaller calibration set (166 samples for the LES calibration set and 120 samples for the stomata calibration set) and the inherently weaker spectral signal associated with stomatal anatomical traits. It is true that the lower predictive ability of some of the traits could amplify the noise in our estimations of trait variability and obscure the relationships between diversity and trait variability, at least in relation to certain functional traits. In any case, we decided to keep the results for stomata in our analyses due to the importance of these traits in leaf strategies, reflecting a gradient mostly related to water use, and because they have been shown to be relevant in tree-tree interactions¹². However, in order to improve clarity about our method and the interpretation of this, we included clarifications about the limitations of our approach based on spectroscopy in the discussion (L451-462).

Supplementary Table 5. Coefficient of determination (R^2) and root mean squared error (RMSE) for each of the eight species included in the study. All calibration samples per species were used to calculate R^2 and RMSE.

Species	SLA	LDMC	C	N	P	SDens	SSize
	R^2						
Castanea henryi	0.92	0.89	0.43	0.90	0.55	0.47	0.20
Nyssa sinensis	0.85	0.94	0.35	0.67	0.39	0.87	0.55
Quercus serrata	0.94	0.85	0.69	0.73	0.54	0.36	0.29
Castanopsis sclerophylla	0.89	0.97	0.54	0.33	0.44	0.67	0.35
Choerospondias axillaris	0.71	0.76	0.32	0.89	0.27	0.84	0.43
Liquidambar formosana	0.95	0.81	0.47	0.93	0.57	0.24	0.11
Sapindus mukorossi	0.95	0.94	0.23	0.83	0.51	0.10	0.16
Triadica sebifera	0.87	0.90	0.41	0.77	0.48	0.39	0.26
	RMSE						
Castanea henryi	11.33	3.17	0.31	0.05	0.81	118.24	1.29
Nyssa sinensis	16.74	3.91	0.22	0.02	0.87	27.38	0.27
Quercus serrata	15.34	3.59	0.01	0.03	0.67	10.18	0.22
Castanopsis sclerophylla	7.85	1.79	0.77	0.03	0.66	35.21	2.79
Choerospondias axillaris	25.50	6.75	0.18	0.01	1.98	26.27	0.84
Liquidambar formosana	19.04	3.32	0.49	0.10	0.23	83.60	0.52
Sapindus mukorossi	14.08	0.61	0.51	0.06	1.38	69.86	0.38
Triadica sebifera	15.41	0.79	0.17	0.04	0.58	205.32	0.35

In addition to this, we checked if our observed patterns for functional traits were similar to patterns when directly analyzing the leaf spectral data, which is an integrative measure of the leaf phenotype. Specifically, leaf reflectance spectra in the domain of solar radiation (350-2500 nm) reflect morphological, physiological, and chemical characteristics related to the plant's growth strategy¹¹. To do so, we calculated the coefficient of variation (CV) in leaf reflectance at the individual (representing intraindividual spectral variability) and population (representing intraspecific spectral variability) level for each wavelength measured from the 400 nm to the 2,500 nm. We can observe that for both intraindividual and intraspecific spectral variability, CV tends to be higher at lower levels of tree species richness (Supplementary Fig. 13, 14). The higher values of CV with lower diversity are especially noticeable in the two short wavelength infrared regions (corresponding to 1550-1800 nm and 2000-2400 nm, respectively¹³). These regions are typically tightly related to leaf functional traits related to the use of resources as the ones included in this study¹⁰. Therefore, we believe that the leaf spectral data supports the patterns

found in our analyses in functional traits. We now refer to this in the discussion (lines L458-460), and include the figures of tree species richness against CV as Supplementary Information (Supplementary Fig. 13, 14).

Supplementary Fig. 13. Coefficient of variation leaf spectra of populations at different tree diversity levels. Coefficient of variation (CV) for leaf spectra of populations of eight different tree species growing along an experimental species richness gradient with mixtures of 1, 2, 4 and 8 tree species (in different colors). Leaf spectra were measured in the range of solar radiation (400–2500 nm) for 12 leaves per individual. The mean leaf spectra of the trees were used to measure CV for each population. Dotted vertical lines represent the limits between the sensors of the spectroradiometer use.

Supplementary Fig. 14. Coefficient of variation leaf spectra of individual trees at different tree diversity levels. Coefficient of variation (CV) for leaf spectra of individual trees of eight different tree species growing along an experimental species richness gradient with mixtures of 1, 2, 4 and 8 tree species (in different colors). Leaf spectra were measured in the range of solar radiation (400–2500 nm) for 12 leaves per individual. The leaf spectra of the trees were used to measure CV for each individual. Dotted vertical lines represent the limits between the sensors of the spectroradiometer use.

5. A defining feature of the BEF-China experiment, distinguishing it from other BEF studies, is its incorporation of substantial environmental heterogeneity. Such heterogeneity, particularly in topography and microsite conditions, is well-documented to drive intraspecific trait variation by influencing individual-level trait expression. While the authors account for slope as an abiotic factor in their model (Line 651), this represents only a partial acknowledgment of the experiment's environmental complexity. For instance, terrain aspect (e.g., north- vs. south-facing slopes) is a critical yet unaddressed variable that directly affects light availability, moisture regimes, and thermal conditions, all of which can induce significant trait plasticity within species. The exclusion of such factors raises concerns about whether the models adequately capture the environmental drivers of trait variation. If unmeasured variables systematically influence trait expression, the study's conclusions about intraspecific variability and its relationship to species richness risk being confounded by residual environmental noise.

>> We agree that environmental heterogeneity within the BEF-China experiment is an important concern, especially regarding topography (see Supplementary Fig. 15). While we controlled for slope, it is true we did not include orientation of the terrain or other factors that could have an effect. Indeed, in a preliminary phase of preparing this manuscript we had already considered orientation as a categorical variable (north, south, east and west). However, most of our trees shared the same orientation (see Supplementary Fig. 26a) and the metrics of intraspecific and intraindividual trait variability did not seem to differ among different orientations (Supplementary Figure 22b, c, d, e). Additionally, we ran the analyses including orientation as a covariate in our linear mixed-effects models and, as orientation was never included among those competing models sharing similar information (according to Burnham & Anderson¹⁴ those with a difference in the Akaike information criterion (AIC) lower than 2 from the model with the lowest AIC), we decided to exclude it from subsequent analyses (see Supplementary Table 8). In order to clarify this, we now mentioned this in the methods (lines L649-651), included a figure in the Supplementary Information describing the distribution of orientations and the values of trait variability for different orientations (Supplementary Fig. 26), and included a table comparing all possible models based on the most complex structure that we use preliminarily to see if it was worth to keep all the covariates in the models used in the analyses (Supplementary Table 8).

Further, it is also true other factors related to environmental heterogeneity for which we do not have data were not included (e.g., microclimate). However, we advocate that: (1) these variables are typically strongly dependent on slope and orientation and (2) by including plot as a random factor we also expect to control for these environmental differences.

Supplementary Fig. 26. Differences in aspect among sampled trees. For each tree sampled, aspect was based on interpolated values of the slope of the terrain obtained from a 5 m resolution digital elevation model (available at <https://data.botanik.uni-halle.de/bef-china/datasets/53>) and used to assign an orientation (east, north, south, west). (a) A circular barplot indicating the number of individuals assigned to each orientation shows that the distribution of orientations is not even, but most of the sampled trees were oriented towards the west. Boxplots indicating differences on (b, d) functional richness (FRic) and (c, e) functional divergence (FDiv) used to assess (b, c) intraindividual and (d, e) intraspecific trait variability in trees sampled for this study. No apparent differences in mean are observed in this figure (but see Supplementary Table 8 addressing the importance of orientation as a covariate in linear mixed-effects models testing the effect of tree species richness on intraspecific and intraindividual trait variability).

Supplementary Table 8. Competing models to identify the drivers of intraspecific and intraindividual trait variability. The competing models for each response variable, defined as those with a difference in Akaike information criterion (Δ AIC) relative to the simplest model (the one with the lowest AIC for each response variable) are highlighted in bold. For each model, information about the estimates of all the included explanatory variables, degrees of freedom, AIC and Δ AIC are included; 'x' indicates whether orientation was included in the model. FRic, functional richness; FDiv, functional divergence; df, degrees of freedom; DBH, diameter at breast height of the tree.

Index	Level	Intercept	Tree richness	Orientation	Slope	DBH	Df	AICc	Δ AICc	
FRic	Intraindividual	7.74				0.11	6	372.56	0.00	
		6.38			0.05	0.11	7	374.82	2.25	
		8.10	-0.36			0.10	7	375.00	2.44	
		9.77					5	375.03	2.47	
		8.18				0.06	6	377.15	4.59	
		6.70	-0.43			0.05	0.11	8	377.24	4.68
		10.16	-0.39				6	377.29	4.72	
		8.42	-0.49			0.06		7	379.28	6.72
		5.88			x		0.10	9	379.65	7.09
		7.04			x			8	381.66	9.09
		6.47	-0.49		x		0.10	10	382.24	9.68
		4.87			x	0.04	0.10	10	382.34	9.77
		5.39			x	0.06		9	384.10	11.54
		7.62	-0.45		x			9	384.17	11.60
		5.42	-0.53		x	0.04	0.10	11	384.97	12.41
5.93	-0.50		x	0.06		10	386.60	14.03		
FDiv	Intraindividual	0.56			0.00		6	-238.02	0.00	
		0.59					5	-237.36	0.67	
		0.56	0.00		0.00		7	-236.69	1.33	
		0.56			0.00	0.00	7	-235.68	2.35	
		0.59	0.00				6	-235.05	2.98	
		0.59				0.00	6	-234.99	3.03	
		0.56	0.00		0.00	0.00	8	-234.26	3.76	
		0.60	0.00			0.00	7	-232.59	5.43	
		0.58			x			8	-232.20	5.83
		0.56			x	0.00		9	-231.26	6.76
		0.59	0.00		x			9	-229.84	8.18
		0.59			x		0.00	9	-229.54	8.48
		0.56	0.00		x	0.00		10	-229.46	8.56
		0.56			x	0.00	0.00	10	-228.60	9.42
		0.59	0.00		x		0.00	10	-227.06	10.96
0.56	0.00		x	0.00	0.00	11	-226.67	11.35		
FRic	Intraspecific	5.68	-0.57				5	325.23	0.00	
		4.98					4	325.83	0.60	
		4.00	-0.75		0.07		6	326.54	1.31	
		3.98			0.04		5	328.19	2.96	
		5.36			x		7	332.35	7.12	
		6.01	-0.56		x		8	333.45	8.22	
		4.02			x	0.04	8	335.22	9.99	
4.00	-0.71		x	0.06	9	335.70	10.47			
FDiv	Intraspecific	0.60			0.00		4	-257.89	0.00	
		0.62					5	-256.56	1.33	
		0.60	0.00				5	-256.27	1.61	
		0.62	0.00		0.00		6	-254.56	3.32	
		0.61			x		7	-250.70	7.19	
		0.64			x	0.00	8	-249.71	8.18	
		0.61	0.00		x		8	-248.86	9.02	
0.63	0.00		x	0.00	9	-247.20	10.69			

Last, in addition to the comments raised by the reviewers, minor changes were made in the manuscript in order to fit the guidelines of the journal and improve the readability of the draft. The most important change has already been described above and includes the removal of the hypotheses at the end of the introduction and description of the main results. Code for the analyses and figures included in this review are also available with the rest of the scripts (<https://doi.org/10.5281/zenodo.14190699>).

REFERENCES

1. Escribano-Rocafort, A. G. *et al.* Intraindividual variation in light-related functional traits: magnitude and structure of leaf trait variability across global scales in *Olea europaea* trees. *Trees - Struct. Funct.* **31**, 1505–1517 (2017).
2. Benavides, R., Scherer-Lorenzen, M. & Valladares, F. The functional trait space of tree species is influenced by the species richness of the canopy and the type of forest. *Oikos* **128**, 1435–1445 (2019).
3. Bittebiere, A. K., Saiz, H. & Mony, C. New insights from multidimensional trait space responses to competition in two clonal plant species. *Funct. Ecol.* **33**, 297–307 (2019).
4. Proß, T. *et al.* Drivers of within-tree leaf trait variation in a tropical planted forest varying in tree species richness. *Basic Appl. Ecol.* **50**, 203–216 (2021).
5. Castro Sánchez-Bermejo, P. *et al.* Tree and mycorrhizal fungal diversity drive intraspecific and intraindividual trait variation in temperate forests: Evidence from a tree diversity experiment. *Funct. Ecol.* **38**, 1089–1103 (2024).
6. Møller, C., March-Salas, M., Kuppler, J., De Frenne, P. & Scheepens, J. F. Intra-individual variation in *Galium odoratum* is affected by experimental drought and shading. *Ann. Bot.* 1–12 (2022) doi:10.1093/aob/mcac148.
7. March-Salas, M., Fandos, G. & Fitze, P. S. Effects of intrinsic environmental predictability on intra-individual and intra-population variability of plant reproductive traits and eco-evolutionary consequences. *Ann. Bot.* **127**, 413–423 (2021).
8. Schnabel, F. *et al.* Tree Diversity Increases Forest Temperature Buffering via Enhancing Canopy Density and Structural Diversity. *Ecol. Lett.* **28**, 1–11 (2025).
9. Ji, F. *et al.* Unveiling the transferability of PLSR models for leaf trait estimation: lessons from a comprehensive analysis with a novel global dataset. *New Phytol.* **243**, 111–131 (2024).
10. Kothari, S. *et al.* Predicting leaf traits across functional groups using reflectance spectroscopy 1 2. (2022) doi:10.1111/nph.18713.
11. Kothari, S. & Schweiger, A. K. Plant spectra as integrative measures of plant phenotypes. *J. Ecol.* **110**, 2536–2554 (2022).
12. Schnabel, F. *et al.* Species richness stabilizes productivity via asynchrony and drought-tolerance diversity in a large-scale tree biodiversity experiment. *Sci. Adv.* **7**, eabk1643 (2021).

13. Li, C., Halitschke, R., Baldwin, I. T. & Schuman, M. C. Evaluating potential of leaf reflectance spectra to monitor plant genetic variation in nature. *Plant Methods* **19**, 108 (2023).
14. Burnham, K. P. & Anderson, D. R. *Multimodel inference: A Practical Information-Theoretic Approach*. *Sociological Methods and Research* (2004).

We thank the reviewers for agreeing again to review our manuscript and for the detailed feedback. We address all questions and comments from Reviewer #2 in detail below, and provide a line-by-line account of the changes made in the substantially revised manuscript. The comments provided by this reviewer led to changes in the analyses and the performance of new analyses that were included in the main body of the paper, as well as other changes that aimed to improve clarity. These revisions did not lead to major changes to the main results or the conclusions of the study. Therefore, we are grateful for these valuable suggestions from the reviewer which helped to strengthen our manuscript. Indeed, the new additional analyses supported the previous conclusions of the study. Please note that the line numbers refer to the clean version without track changes.

Reviewer #1 (Remarks to the Author):

The authors correctly implemented the reviewers' comments. In my opinion the paper can be accepted.

>> We thank Reviewer #1 for the evaluation of our manuscript and for the positive feedback on the revised version.

Reviewer #2 (Remarks to the Author):

Thank you very much for your detailed and thoughtful responses to my earlier comments and questions. While some points have been clarified satisfactorily, several important issues remain unresolved. I would appreciate your further consideration of the following concerns:

1. The introduction currently devotes substantial discussion to how intraspecific trait variability changes with species richness (e.g., the second paragraph). However, your core result (e.g., Fig. 2) primarily addresses how community-level functional diversity changes with richness. There is minimal theoretical framing in the introduction regarding the expectations for results like Fig. 2. This disconnect makes it difficult to interpret the results and understand their theoretical contribution upon reaching them. Crucially, which main figure or table in the results directly addresses the unresolved theoretical questions raised in the second paragraph of the introduction? The current structure creates a sense of disconnection between the introduction and the results.

>> We thank the reviewer for this comment, as it denotes that the parts of the manuscript describing the results from Fig. 2 needed further clarification. The reviewer had the impression that one of our results (shown in Fig. 2) addresses how community-level functional diversity changes with richness. However, it should be noticed that this figure, together with Fig. 1, referred to intraspecific and intraindividual trait variability, as described in the text and in the caption of the figures. Notice that, in our manuscript, intraspecific trait variability refers to the trait differences between individuals within the same population (i.e. conspecifics; see lines L71-73) while intraindividual trait variability refers to the trait differences between leaves within the same tree (see lines L108-110). We believe that this confusion may be caused by the use of functional indices (functional richness (FRic) and functional divergence (FDiv)) to measure intraspecific and intraindividual trait variability. While functional indices are commonly used to assess community functional diversity, they can be used to estimate the trait variability in any assemblage (e.g., population, individual¹). Indeed, the approach of functional indices based on probability densities used in this study aims at measuring trait variability in assemblages at lower levels of biological organization² and the functions to implement it are already prepared for this³. In addition, notice that only Fig. 5 refers to the functional diversity of a community as detailed in the caption.

In order to clarify this, we included changes in the first paragraph of the introduction to specify the use of intraspecific trait variability we make in the manuscript (lines L63-73), in the second paragraph of the introduction to highlight that we aim to test how intraspecific trait variability changes with tree species richness (lines L74-76), in the second paragraph of the results to clarify that we refer to intraspecific and intraindividual trait variability (lines L220, 221, 227, 228) and in the caption of Fig. 2 (line L240). Last, the same icons were consistently used in Fig. 1, 2, 3 and 4 to illustrate intraspecific (between individuals within the same population) and intraindividual (within the same individual) level.

2.Regarding my previous concern about potential sample size limitations (Point #3), your explanations focused on your sampling design philosophy and model robustness. While informative, these do not empirically demonstrate that 20 leaves per species are sufficient to capture the full range of intraspecific trait variation, especially potential multimodality. A more direct approach to address this would involve additional sampling. For instance, selecting a few representative species and measuring a larger number of leaves (e.g., 100 per species) would allow a quantitative assessment of how well the original 20-leaf samples estimate the true intraspecific trait variation within those species.

>> Following the reviewer's concern, we aimed at clarifying the suitability of our sample size and improving transparency when referring to it. First of all, the reviewer is right to point out that explanations on the previous review focused on sampling design philosophy and model robustness. Indeed, we advocated that greater spectral and trait diversity can boost the transferability of predictive models when species are rather similar in the range of their functional traits (as shown in Supplementary Fig. 20). In order to test if 20 samples per species (i.e., 160 samples in total) are sufficient for reaching good predictive ability, we simulated different scenarios of completeness of our training set, to evaluate how the predictive ability of our models was changing. We assessed the coefficient of determination (R^2) of the test set and for each species independently in every case. As shown in the new Supplementary Fig. 26, despite differences among species and traits, most of the species and traits, except for leaf phosphorus content (P), reached stable values of R^2 when the completeness of the training set was 0.5 (meaning that only half of the samples per species were used as part of the training set, resulting in 60 samples). This lower predictive ability for P has been reported in other studies^{4, 5} and it is typically related to the fact that predictive models for target nutrients with overall lower contents in the leaf (such as P) are typically less accurate⁶. We think that, these results, together with the theoretical explanations provided in the previous letter, support that the sample size of the calibration set was sufficient to predict the variability within species in our study. In order to improve transparency, we included this figure in the Supplementary Material, and we made changes in the discussion (lines L493-504), when referring to the limitation of our predictive models. This section of the discussion originally referred to the stomatal traits, which have now been excluded from the manuscript (see response to point 3), but we decided to rewrite it to acknowledge that, while interesting for managing large sample sizes and generating trait datasets that consider the trait variation occurring within species, data analyses based on predictions of leaf functional traits from spectral data (such as leaf phosphorus) can show limitations.

Further, the reviewer suggests that including new samples in our study may be beneficial. We would be willing to do additional measurements in our study site. Nevertheless, in an early stage of our analyses we already tried this and it did not result in better prediction models. Specifically, we aimed at combining our trait and spectral data with the samples collected by Davrinche & Haider⁷ in 2018 in the same experiment, from the same species and using the same spectrometer for acquiring leaf reflectance spectra. However, this increase of the sample size did not improve the accuracy of our models but, instead, seemed to weaken their predictive ability. We suspect that differences in tree age may impede the improvement of the model when adding new samples.

Supplementary Fig. 26. Coefficient of determination (R^2) calculated for the test set and for the samples belonging to each species independently under different scenarios of completeness of the training set. In order to evaluate if the sample size ($n = 160$) was sufficient to predict leaf functional traits effectively from spectral data, we simulated different scenarios of growing completeness of the training set (which was composed of the 75% of the samples of the calibration set), to evaluate changes in the predictive ability of the model in the test set and for the samples of each species independently. Therefore, we fitted predictive models using subsets of the training set representing different proportion of completeness of the training set (10%, 20%, 30%, 40%, 50%, 60%, 70%, 80%, 90% and 100% of samples), and we made sure that the number of samples per species was even in each case. Despite differences among species and traits, most of the traits, except for leaf phosphorus content (P) reached stable values of R^2 when the completeness of the training set was 0.5 (meaning that only half of the samples in the training set were used to train the model). The different pattern in P may explain the lower predictive ability for this trait.

3. The supplementary Table 5 you provided is very helpful and clearly shows that model performance (R^2) is low for many species-trait combinations. You cited two potential reasons: (1) the "smaller calibration set" and (2) the "inherently weaker spectral signal associated with stomatal anatomical traits". Point 1: Acknowledging the small calibration set size implicitly suggests it might be insufficient. Would it not be more advisable to conduct additional measurements to strengthen the calibration for these traits? Point 2: If the spectral signal for stomatal traits is inherently weak, their inclusion in analyses based on predicted values seems problematic, potentially introducing significant noise. A more reasonable approach would be to exclude these traits when calculating community-level functional diversity indices derived from spectral predictions. While I appreciate the ecological importance of stomatal traits ("the importance of these traits in leaf strategies, reflecting a gradient mostly related to water use, and because they have been shown to be relevant in tree-tree interactions"), if they are crucial, the solution should be to directly measure them for the relevant samples and do the functional diversity calculations on these empirical values, rather than relying on potentially unreliable predictions from spectral data.

>> We thank the reviewer for the suggestions about the functional traits. The reviewer mostly refers to stomatal traits and suggests that, while interesting, the low predictive ability could weaken the conclusions of the study. We agree that this is one of the most critical limitations in our study. Therefore, and due to the impossibility of measuring stomatal traits empirically on the leaves that were collected for this study in 2023, we decided to exclude these functional traits from our analyses. While we acknowledge the ecological importance of stomatal traits as indicators of water-related strategies in leaves, especially in the context of diversity, we decided to focus on functional traits of the leaf economics spectrum (LES), which represent an important part of tree strategy and economy⁸. This resulted in small changes in the analyses, that have been reflected along the whole results section (e.g. changes in the variance explained by each PC axis in lines L184-190, slight changes in P values in lines L224, 230, 232, 235, 236, 237, 289, 290, 291, 292, 293, 333 and 342) and in Fig. 1, 2, 4 and 5. Additionally, Supplementary figures and tables related to results were also modified (Supplementary Fig. 2, 3, 4, 9, 10, 13, 15, 21, 22, 23, 24, 25, 29 and Supplementary Table 1, 2, 5, 9).

After removing the stomatal traits, the results remained qualitatively similar. For instance, the principal components in Fig. 1 changed slightly but, as shown in the previous version of the manuscript, they were still related to leaf morphology (leaves with high specific leaf area (SLA) and low leaf dry matter content (LDMC) and leaf carbon content (C) that we now decided to describe as thinner versus leaves with high LDMC and low SLA that we describe as thicker) and nutrition status (low content of leaf nutrients versus high content of leaf nutrients). In addition,

the changes on intraspecific and intraindividual trait variability followed similar patterns as shown before (see Fig. 1, 2, 4) and the results for the functional diversity of tree communities reflected the same pattern as before (see Fig. 5). This similarity in the results resulted in few changes in the discussion. Main changes relate to the fourth paragraph of the discussion about intraindividual variability (lines 412-442), in which part of the discussion related to stomatal traits in the previous version. However, due to the exclusion of stomatal traits from the study, and the inclusion of new analyses on spectral traits (see point 4 about new analyses on leaf reflectance data), the discussion was modified in lines L424-442. Also, despite these changes in the discussion, the main conclusions of the study remained untouched.

Further, the exclusion of stomatal traits from our analyses implied changes in the methods (e.g. changes in lines L556-565 about the calibration set, changes in 587-601 about the laboratory analyses) and the methodological Supplementary Materials (Supplementary Fig. 19, 27).

4.I appreciate the inclusion of CV analysis of spectral data (Supplementary Figs. 13 & 14) as supporting evidence. However, the figures show considerable variation. To convincingly demonstrate that the spectral results align with the main trait-based findings (i.e., lower variability at low richness), formal statistical testing of the relationship between species richness and spectral CV is necessary. Furthermore, to fully leverage this alternative approach, it would be highly informative to present the core results (analogous to your main Fig. 2 and 3) derived solely from the spectral data within the main manuscript.

>> We thank the reviewer for the idea of including analyses on the spectral data. Accordingly, we decided to test changes in intraindividual and intraspecific trait variability using the spectral data. In order to be consistent with the other analyses, we aimed at identifying principal components that summarize the leaf reflectance spectrum (as in Li et al.⁹) and used them to estimate spectral trait variability using the same approach that we used for functional traits. That is why, we included a new section in the methods about the measurements of spectra variability (lines L671-693) and new Supplementary Materials (Supplementary Fig. 6, 7) that contribute to detail the methods for these analyses

As shown in Fig. 3 (a new figure included in the main text showing the main results for the analyses on the spectral data), these results support the main conclusion that intraspecific and intraindividual phenotypic variability decrease with tree species richness. Changes in the results have been included in lines L250-261 and Fig. 3. Further, additional results related to these analyses are included in the new Supplementary Fig. 5, 7.

Interestingly, these analyses showed prominent results in the case of the intraindividual variability and showed that the spectral regions responding to tree species richness were different from those related to traits and were located in the range of the visible light (VIS; 400-700 nm) and the “red edge” transition (680-750 nm). These regions are strongly linked to leaf photosynthetic and protective pigments, such as chlorophyll or carotenoids (as shown in Supplementary Fig 14), providing new insights into the discussion of our results. This led to important changes in the fourth paragraph of the discussion, where we discussed the causes of the changes in intraindividual variability as reflected by the leaf reflectance spectra (lines L424-442).

To sum up, we believe that our manuscript benefited from these additional analyses that give support to the main conclusions of the manuscript.

Fig. 3. Spectral segmentation and effect of tree species richness on the variability of spectral components at the intraspecific and intraindividual level. (a) Fragmentation of leaf reflectance spectrum into segments holding one identified principal component using a Hierarchical Spectral Clustering with Parallel Analyses (HPS-CA; Supplementary Fig. 6, 7) on 4568 leaf reflectance spectra collected from trees growing along an experimental species richness gradient with mixtures of 1, 2, 4 and 8 tree species. Each segment is represented in a different color and the line represents the mean reflectance measured at different wavelengths (from 400 to 2500nm) in our study. Regression estimates from linear mixed-effects to study intraindividual and intraspecific spectral variability of the principal components associated to the identified segments **(b)** show a significant decrease of intraindividual FRic with tree species richness in 8 principal components (the ones associated with segments 15, 18, 23, 24, 30, 31, 35, and 36; $N = 381$) and **(c)** a significant decrease of intraspecific FRic with tree species richness in one principal component (the one associated with the segments 18; $N = 64$). Colors of the regression estimates represent the significance as determined by a likelihood ratio test (black $P < 0.05$, grey $P > 0.05$), while colors on the axis correspond to the segments in **(a)**. **(d)** Correlation between principal components associated to identified segments and leaf functional traits used in the study. Each circle represents a correlation, with size representing the R^2 and the color indicating the direction of change (white, negative; black, positive). Segment numbers are derived from the HSC-PA process shown in Supplementary Fig. 7.

Supplementary Fig. 5. Regression coefficients for the effects of tree species richness on the intraspecific and intraindividual spectral variability on 29 principal components associated with segments of the leaf reflectance spectrum. Regression coefficients for the effects of tree species richness on the intraspecific and intraindividual variability on seven leaf functional traits and two main axes of leaf trait variability. The effects of tree species richness on intraspecific and intraindividual variability were studied for 29 principal components associated with segments of the leaf reflectance spectrum (see Fig. 3 for details about the segments). Colors represent the significance as determined by a likelihood ratio test (red $p < 0.05$, pink $0.01 < p < 0.05$, grey $p > 0.05$).

Supplementary Fig. 6. Conceptual framework for the segmentation of the leaf reflectance spectrum using the Hierarchical Spectral Clustering with Parallel Analysis (HSC-PA). (a) By using a Horn's parallel analysis on all the wavelengths of the leaf reflectance spectrum (2101 wavelengths), the number of retained principal components was assessed. A principal component was retained when its associated eigenvalue was higher than 1. If the number of principal components retained was higher than one, (b) then the wavelengths were divided into two groups (i.e. segments) using spectral clustering. The resulting segments were then used to repeat this process, always dividing the wavelengths associated to each segment into two groups until (c) a segment retaining one unique principal component was identified. Annotations in grey indicate the functions and the package used for the steps of the analysis. Wavelengths associated to each segment are colored in red.

Supplementary Fig. 7. Segmentation of leaf reflectance spectrum obtained using Hierarchical Spectral Clustering with Parallel Analysis (HSC-PA). Segments are colored in red. From the center, each concentric circle represents the partition of one segment into two. Segmentation process was performed by using HSC-PA as illustrated in Supplementary Fig. 6.

Supplementary Fig. 14. Description associations between different regions of the leaf reflectance spectrum and biochemical and structural components of leaves (adapted from Li et al.⁶⁴). The main regions represented in the leaf reflectance spectrum include the visible range (VIS) and the red edge region (400-750 nm), the near infrared region (NIR; 800-1300 nm) and the short-wavelength infrared regions (SWIR), composed by two regions (SWIR1 and SWIR2) that are separated by water absorption bands. The line represents the mean reflectance measured at different wavelengths (from 400 to 2500nm) in our study, while colors represent the different segments illustrated in Fig. 3.

5. At last, thank you for addressing the aspect of orientation and confirming its lack of systematic influence. However, an additional concern arises: is there a systematic difference in the CV of environmental factors across plots of different species richness levels? For example, what pattern emerges if the y-axis in a figure like Fig. 2 represented the CV of key abiotic factors (e.g., soil moisture, light availability) instead of trait diversity? Heterogeneity in abiotic environmental conditions is a known driver of both intraspecific trait variation and functional diversity, and its potential correlation with the manipulated species richness gradient needs examination.

>> Following the reviewer's concern, we decided to test how the variance in slope and aspect changed along the diversity gradient. Therefore, we quantified the variability in slope and aspect as the variance of all the pixels that fall within each sampled plot in a 5 m resolution digital elevation model, available at the experiments' data base (<https://data.botanik.uni-halle.de/bef-china/datasets/53>). In order to address their changes in response to diversity, we used a linear model with tree species richness as the predictor and variance as the response variable. The results showed lack of significance ($P = 0.29$ in the case of slope and $P = 0.73$ in the case of aspect; $n = 32$) and very weak correlations ($R^2 = 0.04$ in the case of slope and $R^2 < 0.01$ in the case of aspect; Review Fig. 1). We therefore conclude that heterogeneity in these abiotic factors is not systematically related to the species richness gradient. This supports our interpretation that the patterns observed in our study primarily result from tree species richness itself.

Apart from these environmental variables, the reviewer mentioned others, such as soil moisture and light availability, but these have not been included because:

(1) Many environmental variables related to soil, light, etc. have already been shown to change in response to diversity itself^{10, 11, 12}. Therefore, we are worried that, when evaluating how their variability changes along the diversity gradient, the results are blurred by the diversity effects.

(2) Data availability limits our ability to quantify the variance of these environmental variables at the plot level. Mean plot values have been measured in the experimental site in several studies and, while we agree that the variance remains of interest, sufficient data to estimate variance of environmental variables reliably at the plot level is scarce.

However, we believe that the main environmental factors of interest, such as soil nutrient content or light exposure, are most likely influenced by the variable topography (slope and aspect) in our study site.

Review Fig. 1. Correlation between variance in topography (slope and spect) and tree species richness in the studied plots. Dashed lines indicate lack of significance ($P > 0.05$) and grey dashed areas indicate confidence intervals.

REFERENCES

1. Palacio, F. X., Ottaviani, G., Mammola, S., Graco-Roza, C., de Bello, F., & Carmona, C. P. (2025). Integrating intraspecific trait variability in functional diversity: An overview of methods and a guide for ecologists. *Ecological Monographs*, 95(2), e70024.
2. Carmona, C. P., de Bello, F., Mason, N. W. H, Lepš, J. Traits without borders: integrating functional diversity across scales. *Trends Ecol. Evol.* **31**, 382–394 (2016).
3. Carmona, C. P., de Bello, F., Mason, N. W. H & Lepš, J. Trait probability density (TPD): measuring functional diversity across scales based on TPD with R. *Ecology* **100**, e02876 (2019).
4. Proß, T. *et al.* Drivers of within-tree leaf trait variation in a tropical planted forest varying in tree species richness. *Basic Appl. Ecol.* **50**, 203–216 (2021).
5. Proß, T., Haider, S., Auge, H. & Bruelheide, H. Leaf trait variation within individuals mediates the relationship between tree species richness and productivity. *Oikos*, e10255 (2023).

6. Murguzur, F. J. A. *et al.* Towards a global arctic-alpine model for Near-infrared reflectance spectroscopy (NIRS) predictions of foliar nitrogen, phosphorus and carbon content. *Scientific Reports* **9**, 8259 (2019).
7. Davrinche, A. & Haider, S. Intra-specific leaf trait responses to species richness at two different local scales. *Basic Appl. Ecol.* **50**, 20–32 (2021).
8. Wright, I. J. *et al.* The worldwide leaf economics spectrum. *Nature* **428**, 821–827 (2004).
9. Li, C., Czyż, E. A., Ray, R., Halitschke, R., Baldwin, I. T., Schaepman, M. E. & Schuman, M. C. Association of leaf spectral variation with functional genetic variants. *bioRxiv* (2018).
10. Li, Y *et al.* Early positive effects of tree species richness on soil organic carbon accumulation in a large-scale forest biodiversity experiment. *J. Plant. Ecol.*, **12**, 882–893 (2019).
11. Hildebrand, M., Perles-Garcia, M. D., Kunz, M., Härdtle, W., von Oheimb, G. & Fichtner, A. Tree-tree interactions and crown complementarity: The role of functional diversity and branch traits for canopy packing. *Basic Appl. Ecol.* **50**, 217–227 (2021).
12. Schnabel, F. *et al.* Tree Diversity Increases Forest Temperature Buffering via Enhancing Canopy Density and Structural Diversity. *Ecol. Lett.* **28**, 1–11 (2025).